# Breaking the Batch Barrier (B3) of Contrastive Learning via Smart Batch Mining

**Raghuveer Thirukovalluru**[†]    **Rui Meng**[‡*]    **Ye Liu**[‡]    **Karthikeyan K**[†]    **Mingyi Su**[¶]

**Ping Nie** [§]    **Semih Yavuz**[‡]    **Yingbo Zhou**[‡]    **Wenhu Chen**[¶]    **Bhuwan Dhingra**[†]

Duke University[†]    Salesforce AI Research[‡]    Independent[§]    University of Waterloo[¶]

rt195@duke.edu

## Abstract

Contrastive learning (CL) is a prevalent technique for training embedding models, which pulls semantically similar examples (positives) closer in the representation space while pushing dissimilar ones (negatives) further apart. A key source of negatives are "in-batch" examples, i.e., positives from other examples in the batch. Effectiveness of such models is hence strongly influenced by the size and quality of training batches. In this work, we propose *Breaking the Batch Barrier* (B3), a novel batch construction strategy designed to curate high-quality batches for CL. Our approach begins by using a pretrained teacher embedding model to rank all examples in the dataset, from which a sparse similarity graph is constructed. A community detection algorithm is then applied to this graph to identify clusters of examples that serve as strong negatives for one another. The clusters are then used to construct batches that are rich in in-batch negatives. Empirical results on the MMEB multimodal embedding benchmark (36 tasks) demonstrate that our method sets a new state of the art, outperforming previous best methods by +1.3 and +2.9 points at the 7B and 2B model scales, respectively. Notably, models trained with B3 surpass existing state-of-the-art results even with a batch size as small as 64, which is 4–16× smaller than that required by other methods. Moreover, experiments show that B3 generalizes well across domains and tasks, maintaining strong performance even when trained with considerably weaker teachers.

## 1   Introduction

Contrastive Learning (CL) has emerged as the dominant approach for training embedding models [5, 2, 8]. It typically operates on data in the form of (query, positive) pairs [29], where the objective is to minimize the distance between the query's representation and that of its positive counterpart. To foster more discriminative representations, CL also incorporates negative examples – instances which, for a given query, are not its designated positive – by increasing their representational distance from the query. In practice, other examples within the same batch serve as readily available in-batch negatives [9]. Furthermore, to significantly enhance the learning signal, "hard negatives" are often explicitly mined and integrated into the training process [5, 11]. These hard negatives are particularly challenging examples that share superficial features with the query or its positive, making them easily confusable with the true positive, yet are semantically distinct.

---

[*]Now at Google;    Code and Models: https://github.com/raghavlite/B3

39th Conference on Neural Information Processing Systems (NeurIPS 2025).

Recent text-only embedding models, such as NV-Embed [11] and SFR-Embedding [18], have achieved state-of-the-art results through the use of hard negative mining from the training dataset. These methods typically employ a pretrained teacher model to rank potential negatives relative to a given query, subsequently selecting a small set (e.g., up to ten) of top-ranked candidates as hard negatives. While incorporating multiple hard negatives can enhance model performance, it also substantially increases training cost and duration (e.g. using ten hard negatives is approximately 4x training time compared to using one hard negative). This performance-cost trade-off, while manageable for text-only models, becomes computationally prohibitive in multimodal contexts. In such settings, positive examples often include high-resolution images, significantly amplifying the processing overhead associated with each additional hard negative [30]. Moreover, contrastive learning relies on large datasets, which are already hard to fully use in multimodal training [14, 24]. Adding many hard negatives makes this even more demanding, limiting scalability.

Consequently, many recent efforts in multimodal embedding learning have largely avoided explicit hard negative mining across the entire training dataset. For instance, VLM2Vec [8] repurposes data from diverse tasks – such as retrieval, classification, and VQA – for contrastive training but omits dedicated hard negative sampling. MegaPairs [38] and mmE5 [1] focus on generating synthetic contrastive pairs, prioritizing data augmentation over mining challenging negatives. Other methods, including LLaVE [10] and UniLM [6], do incorporate negative sampling strategies, but restrict this to resampling examples within the current batch as negatives. While these approaches have proven effective to varying degrees, they often do not exploit the potentially stronger negative signals residing outside the immediate batch, thereby overlooking a rich source of contrastive supervision.

This work addresses the aforementioned gap by constructing batches from the full dataset where examples within each batch serve as strong negatives for one another. This approach aims to eliminate the need for separately mined hard negatives, significantly reducing computational cost. Our proposed batch mining method, B3, leverages teacher-based ranking but revolutionizes batch construction. Unlike prior methods that add individually mined hard negatives to IID sampled batches, B3 uses graph-based community detection to cluster examples which are mutually strong negatives of each other. Batches are then strategically formed by sampling from these cohesive communities ensuring strong in-batch negatives. This facilitates strong training signals without the need for extra negatives.

Our contributions are summarized as follows:

- We introduce B3, a novel batch mining strategy that sets a new state-of-the-art on the MMEB benchmark, surpassing strong baselines by 1.3 and 2.9 points at the 7B and 2B scales resp.
- We provide theoretical justification for the design of B3.
- We conduct a comprehensive ablation study to evaluate the individual components of B3 and highlight the significant impact of the batch mining module.
- B3 achieves sota performance at 2B scale despite training at a much smaller batch size of 64.
- B3 beats random batching baseline with strong hard negatives using only half the compute.
- B3 works effectively with weak teachers and generalizes across domains.

## 2 Related Work

This section reviews prior work on improving negative sampling for contrastive learning, strategies for batch mining, and recent advances in multimodal embedding methods.

### 2.1 Mining Better Negatives

E5 [31] leveraged GPT-4 to construct contrastive learning (CL) datasets with diverse-length (anchor, positive, negative) tuples, which were then used to fine-tune the Mistral 7B model. SumCSE [28] used compositional transformations to create negatives. Gecko [12] employed a large LLM to generate queries from existing passages and subsequently relabeled positives and negatives for these queries. In contrast, our approach treats positives from other examples as potential negatives for a given query.

Orthogonal to this, several recent studies, including SFR-Embedding [18, 17], NV-Embed [11] and NV-Retriever [20], incorporate diverse annotated datasets — including clustering and classification — in addition to retrieval data, achieving notable gains on MTEB. A common technique in these methods involves using a pretrained teacher model to rank all positive examples from other queries in

the dataset relative to a given query, selecting the top-ranked instances as hard negatives. Drawing inspiration from these studies, our approach also employs a teacher model to assess relationships between examples. However, instead of directly selecting and adding hard negatives, we leverage this ranking information to construct high-quality batches where the constituent examples inherently serve as strong negatives for one another during contrastive training.

## 2.2 Batch Mining for Contrastive Learning

Several studies have explored improving contrastive models by optimizing the construction of training batches. NGAME [4] forms batches by clustering data points and merging clusters, targeting extreme multi-label classification tasks. GCBS [25] formulates batch mining as a matrix bandwidth minimization problem on adjacency graphs derived from intermediate model checkpoints. BatchSampler [33] employs random walks over adjacency graphs to sample informative batches. However, these methods do not account for the impact of false negatives or the detrimental effects of excessive number of hard negatives on the batch construction process, particularly in large-scale datasets. More recently, Morris and Rush [21] employed clustering to construct batches containing stronger negatives; however, their approach is limited to text-only benchmarks and smaller-scale models.

## 2.3 Improving Multimodal Embeddings

**CLIP Based**: CLIP [24] trained separate encoders for images and captions using 400M (image, caption) pairs. DreamLip [37] built on this by introducing additional losses to align subcaption embeddings with image patch embeddings. MagicLens [36] utilized naturally occurring image pairs and generated text describing their differences, using these as contrastive instructions for training multimodal embeddings. However, these LIP-style models employed separate encoders for image and text. In contrast, recent studies showed that vision-language models (VLMs) with early fusion of image and text features achieved better performance on multimodal embedding tasks.

**VLM-based methods**: E5-V [7] employs EOL-style prompts (e.g., "Text/Image means in one word:") on text-only data to train a VLM for embeddings. VLM2Vec [8] is trained on pairwise data curated from diverse tasks—including retrieval, classification, visual question answering (VQA), and grounding—but does not incorporate hard negatives during training. MegaPairs [38], similar to MagicLens, leverages a vision-language model to generate questions linking semantically related image pairs. LLaVE [10] introduces a weighting mechanism over in-batch negatives, assigning higher weights to negatives with larger logits relative to a query, thereby prioritizing them during contrastive training. mmE5 [1] synthesizes multimodal triplets (query, positive, negative) involving diverse image-text combinations for training, drawing inspiration from E5. UniLM [6] enhances embedding quality by distilling knowledge from text-only models and resampling in-batch negatives as hard negatives. Despite their innovations, these MLLM-based approaches do not fully exploit the potential of strong negatives that can be systematically mined from the training dataset.

## 3 Methodology

Our proposed methodology comprises three key components: (1) a novel batch selection mechanism, which forms the core of our approach; (2) an optional negative sampling strategy tailored to these curated batches; and (3) enhanced prompting techniques to guide the embedding model.

### 3.1 Batch Selection

In contrastive learning, performance is critically influenced by both batch size and the quality of negatives within each batch. Addressing this, our work introduces a novel batch mining strategy that leverages teacher-model rankings to strategically compose batches. The aim is to ensure that the examples within each batch inherently act as strong negatives for each other, thus providing rich contrastive signals efficiently. The following subsections will elaborate on the derivation and components of this methodology.

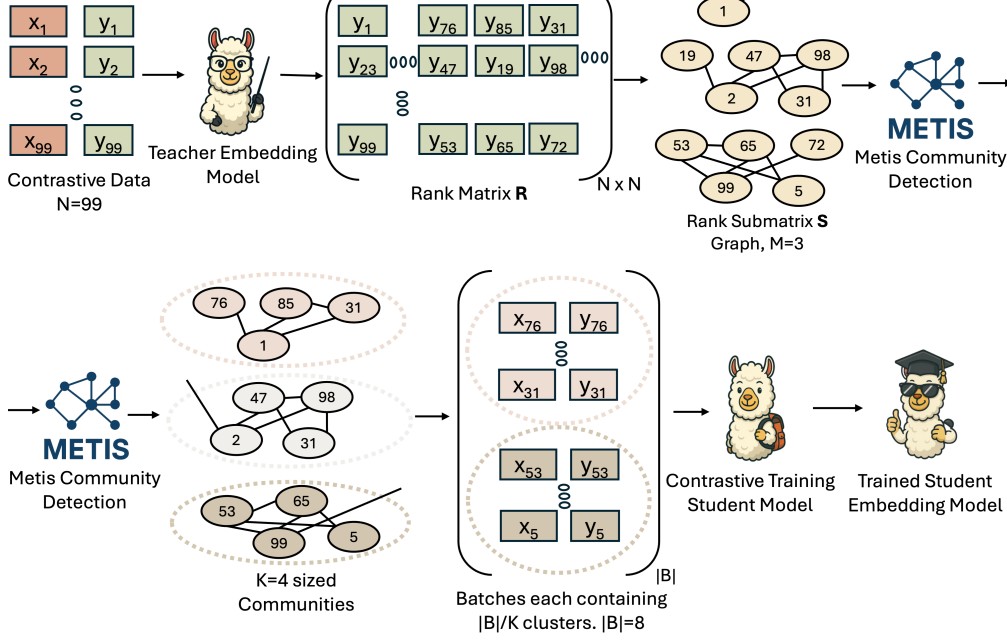

Figure 1: The batch mining mechanism of B3. Initially, a teacher model generates a rank matrix $R$ over the training set, indicating potential negative relationships. From these rankings (specifically ranks in the range $[p : p + m]$ for each query), a undirected sparse preference graph $S$ is constructed. Then, METIS clustering is applied to identify communities of mutually strong negatives. Finally, diverse training batches of size |B| are formed by sampling examples from $|B|/K$ distinct communities.

### 3.1.1 Batch Mining Algorithm

Our algorithm starts by using a trained teacher embedding model to rank all positives in the training set w.r.t a given query, $x_i$. Let $R \in \mathbb{N}^{N \times N}$ be the **rank matrix**, where each row $R_i = R_{i,:}$ is a permutation of the set $\{1, 2, \ldots, N\}$. For each row $i$, the entries represent the ordered ranks of the $N$ examples, sorted from highest to lowest relevance (or similarity) of the positive $y_i$ with respect to query $x_i$ according to the teacher model. The highest scoring targets for each query when used as in-batch negatives adversely affect performance due to the presence of False Negatives (targets which are semantically equivalent/closer to golden target for the query and cannot be used as a negative). To filter out these false negatives, we use a simple rank based thresholding as proposed in NV-Retriever [20].We exclude the top $p$ ranks and only use the next $m$ ranks to sample in the batch mining procedure.

Our final algorithm uses only the specified columns, i.e., $S = R[:, p : p + m]$. This submatrix represents the adjacency structure of a sparse directed graph. In constructing the final undirected graph $S$, we retain only the bidirectional edges. As can be seen, the edges in the graph denote preference for same batch (strong in-batch negatives in our case). We now apply METIS community algorithm [16] to identify clusters of size $K$ from this graph. METIS is a minimum cut algorithm - maximizing the number of edges within communities and minimizing the edges across across communities. Given the smaller value of $m$, METIS is fairly fast and approximately only requires $\mathcal{O}(n)$ runtime.

Given the clusters of size $K$, we collect $|B|/K$ random such clusters to construct a batch. Note that contrastive learning needs high batch size $|B|$ [2, 5]. It is also simple enough to derive the same from Eq. 2, that higher batch sizes minimize this difference. The algorithm is depicted in Fig. 1.

### 3.1.2 Theoretical Justification of the Proposed Algorithm

Taking inspiration from Sachidananda et al. [25], we use the global contrastive loss as the reference to derive a theoretical justification for our algorithm. The global InfoNCE loss is defined below, with the summation taken over all examples in the dataset. The temperature term is omitted for brevity.

$$\mathcal{L}^{Global} = \sum_{i=1}^{N} -\log \frac{exp(x_i^T y_i)}{\sum_{j=1}^{N} exp(x_i^T y_j)} \tag{1}$$

where $x_i$ is a query and $y_i$ is the corresponding labeled target, N is the size of the dataset. Our goal is to build a batch $B$ that can approximate the global loss on the batch level InfoNCE loss as follows.

$$\min_{B} \quad \mathcal{L}^{Global} - \mathcal{L}^{B} \tag{2}$$

$$\text{where} \quad \mathcal{L}^{B} = \sum_{i=1}^{N} -\log \frac{exp(x_i^T y_i)}{\sum_{j \in B_i} exp(x_i^T y_j)} \tag{3}$$

**Theorem 1.** *The difference between global and batch loss terms is upper-bounded as follows:*

$$\mathcal{L}^{Global} - \mathcal{L}^{B} \leq \sum_{i=1}^{N} \log \left( \frac{N}{K} \frac{H_i^K}{H_{B_i,i}^K} \right) \tag{4}$$

*where $H_i^K$ and $H_{B_i,i}^K$ denote the sum of the top $K$ exponent terms in the denominator for the global and batch loss components, respectively, for each query $x_i$.*

The proof of the theorem is presented in §E. Since the bound holds for all values of $K$, we henceforth let $K$ represent the number of top in-batch elements that encompass all strong negatives for query $x_i$ within the batch $B_i$. The two mains things that we note from this bound: (1) A higher value of $H_{B_i,i}^K$ makes the bound tighter; and (2) A higher value of $K$ makes the bound tighter.

**Theorem 2** ([26]). *Loss of an embedding model on downstream tasks is bounded as follows:*

$$\mathcal{L}_{sup}(\hat{f}) \leq \alpha L_{un}^{\neq}(f) + \beta s(f) + \eta \, Gen_M \tag{5}$$

*where $s(f)$ is the a measure of intra-class covariance and $\beta$ is a term that increases with more strong negatives for a query.*

Refer to [26] for a formal proof. Both intra-class covariance and $\beta$ increase with a higher number of strong negatives, i.e. $K$. This theorem calls for avoiding picking very large values of $K$. The claim is also empirically corroborated by SFR-Embedding [19], where using high number of hard negatives resulted in inferior performance. Given the opposing trends for $K$ observed in both theorems, it is most appropriate to select $K$ empirically. Note that $K$ is limited by the batch size $|B|$.

The bound in Eq: 4 should be optimized for all stages during training. Effectively, given a $K$, $H_{B_i,i}^K$ needs to be simultaneously maximized $\forall i, B_i$ i.e.

$$\max_{B_1,..B_i..,B_{N/|B|}} \quad \prod_{i=1}^{N} H_{B_i,i}^K \tag{6}$$

Algorithm proposed in §3.1.1 does optimize for this. Each cluster is set to be size $K$. Note that the number of edges within each cluster are maximized. This maximizes the $H_{B_i,i}^K$ term as edges denote preference for strong negatives. Our algorithm jointly optimizes batch composition to ensure that groups of mutually preferred negative examples are co-located within the same batch. It is important to note that one could, in principle, increase $H_{B_i,i}^K$ by directly adding hard negatives while retaining random batches; However, this approach would be extremely computationally expensive.

### 3.2 Unified Hard-Negative Mining (Optional)

While the mining algorithm in §3.1 is expected to build batches whose examples are strong in-batch negatives of each other, it might turn out that some examples are just not possible to put together with its preferred rank examples due to the collective optimization. Additional mined hard negatives might be important for such cases. In this work, we introduce an enhancement over the approach of randomly sampling $h$ additional negatives per query from $S$, as proposed in SFR-Embedding.

Note that each additional hard-negative is contrasted against all queries in the batch. Hence, rather than naively using $S[i,:]$ to sample hard negatives for query $x_i$, we come up with an aggregated

strategy to account for preferences of all queries in the batch mined in §3.1.1. A unified probability distribution $\Pr(j)$ is created such that $\Pr(j) \propto \text{Count}(S_B, j)$. $\text{Count}(S_B, j)$ is the number of times item $j$ appeared in the rows of $S$ corresponding to the items/queries in batch $B$. This unified distribution is used to sample $h$ hard negatives per examples for all examples in the batch.

### 3.3 Improved Representation Prompts

Recent work on multi-modal embeddings has utilized instructions to encode queries, but has typically represented positive examples directly, without using instruction prompts [8, 10]. This mismatch introduces inconsistencies in how different types of positives are represented. For example, a positive such as "aeroplane" in ImageNet classification requires a fundamentally different representation compared to a detailed image caption from MSCOCO, such as "Skateboarder performing a stunt in mid air." Incorporating instruction prompts for positives is also expected to help decouple stylistically diverse tasks during training. Hence, we design positive target prompts for all datasets in §I.

### 3.4 Overall Methodology

We introduce two variants of our methodology: B3 and B3++. B3 serves as the core approach, integrating the components from §3.1 and §3.3, and delivers strong performance across benchmarks. B3++ builds on B3 by additionally incorporating hard negatives as described in §3.2, offering further gains when computational resources allow. While B3++ provides enhanced performance, B3 remains highly effective and is well-suited for resource-constrained settings.

## 4 Experiments

**Training Set** We train our models on the MMEB [8] training set. We train for 2000 steps (~2 epochs) with a batch size of 1024 unless specified.

**Evaluation Set** We evaluate our methods on the MMEB [8] benchmark. MMEB benchmark contains 36 distinct tasks spanning four diverse categories — Retrieval (12), Classification (10), Visual Question Answering (10), Grounding (4). It contains 20 In-Domain and 16 Out-of-Domain tasks. Evaluation metric for all datasets is accuracy. Average accuracy of all datasets is reported.

**Methods Compared**: We primarily evaluate the proposed methods—B3++ and B3—across various batch sizes. As a baseline, we also include `Random Batches`, which employs random batch selection. Additionally, we compare all methods with publicly accessible training methodology prior to the submission deadline.

**Implementation**: We test our methodology using two different VLMs - `Qwen2-VL-7B-Instruct`, `InternVL3-8B` with both 7B and 2B variants. We use $m = 100$ following SFR-Embedding and NV-Retriever. For $p$, we tuned this value on heldout portions of the train set. We used $p = 30$ for retrieval and grounding tasks and $p = 70$ for VQA tasks. For classification tasks, we just filter out the golden label from the rank list. We use 5 hard-negatives $h = 5$ for B3++ and no hard negatives in B3. All models in this work are trained using LoRA with a rank of 8. Unless mentioned, models are trained for 2k steps with peak learning rate of 1e-4 and warmup of 10%. Temperature used was 0.02.

**Teacher Models**: To effectively capture the gains from our batch mining, we use teacher models - with the same scale, trained on the same data, without hard negatives. In our analysis, we used `VLM2Vec` (Qwen2-2B) to train 2B student models, and `VLM2Vec` (Qwen2-7B) to train 7B student models. The batch mining was performed at a task level for each of the 20 MMEB training datasets.

### 4.1 Main Results

Table 1 presents the performance of our models. At both the 2B and 7B model scales, our proposed methodology outperforms existing approaches. Specifically, B3++ (Qwen2-2B) surpasses the next best model by a substantial margin of 2.9 points, while B3++ (Qwen2-7B) achieves a notable improvement of 1.3 points, averaged across 36 tasks. While other models have significant data and modeling level differences, a more natural baseline for comparison is our teacher model, VLMVec. **B3++ beats VLM2Vec by 6.2 points (Qwen2-7B) and 8.8 points (Qwen2-2B)**.

Table 1: Main Table: B3++ achieves notable performance gains over the current best models on MMEB. B3++ outperforms `VLM2Vec` - the teacher model and a more relevant baseline—by 6.2 points (for Qwen-7B) and 8.8 points (for Qwen-2B).

| Model | Ret. | Cla. | VQA | Gro. | ID | OOD | Avg. |
|---|---|---|---|---|---|---|---|
| #Tasks | 12 | 10 | 10 | 4 | 20 | 16 | 36 |
| *LIP Style Baselines* | | | | | | | |
| CLIP [24] | 42.8 | 9.1 | 53.0 | 51.8 | 37.1 | 38.7 | 37.8 |
| BLIP2 [13] | 27.0 | 4.2 | 33.9 | 47.0 | 25.3 | 25.1 | 25.2 |
| SigLIP [34] | 40.3 | 8.4 | 31.6 | 59.5 | 32.3 | 38.0 | 34.8 |
| OpenCLIP [3] | 47.8 | 10.9 | 52.3 | 53.3 | 39.3 | 40.2 | 39.7 |
| UniIR (BLIP$_{FF}$) [32] | 42.1 | 15.0 | 60.1 | 62.2 | 44.7 | 40.4 | 42.8 |
| UniIR (CLIP$_{SF}$) [32] | 44.3 | 16.2 | 61.8 | 65.3 | 47.1 | 41.7 | 44.7 |
| MagicLens [36] | 38.8 | 8.3 | 35.4 | 26.0 | 31.0 | 23.7 | 27.8 |
| *~2B VLM Models (Trained on MMEB)* | | | | | | | |
| VLM2Vec (Qwen2-2B) [8] | 65.4 | 59.0 | 49.4 | 73.4 | 0.0 | 0.0 | 59.3 |
| UniME (Phi-3.5-V) [6] | 64.5 | 54.8 | 55.9 | 81.8 | 68.2 | 52.7 | 64.2 |
| LLaVE (Aquila-VL-2B) [10] | 65.2 | 62.1 | 60.2 | 84.9 | 69.4 | 59.8 | 65.2 |
| B3 (InternVL3-2B) **(Ours)** | 69.0 | 62.6 | **64.0** | **86.9** | **73.5** | 60.8 | 67.8 |
| B3++ (Qwen2-2B) **(Ours)** | **70.9** | **67.0** | 61.2 | 79.9 | 72.1 | **63.1** | **68.1** (+2.9) |
| *>7B VLM Models (Trained on MMEB)* | | | | | | | |
| VLM2Vec (Qwen2-7B) [8] | 69.9 | 62.6 | 57.8 | 81.7 | 72.2 | 57.8 | 65.8 |
| MMRet (Llava-Next-7B) [38] | 69.9 | 56 | 57.4 | 83.6 | 68 | 59.1 | 64.1 |
| mmE5 (Llama-3.2-11B) [1] | 70.9 | 67.6 | 62.8 | 89.7 | 72.3 | 66.7 | 69.8 |
| LLaVE (Llava-OV-7B) [10] | 70.9 | 65.7 | 65.4 | **91.9** | 75.0 | 64.4 | 70.3 |
| UniME (LLaVA-OneVision-7B) [6] | 70.5 | 66.8 | 66.6 | 90.9 | 74.6 | 65.8 | 70.7 |
| B3 (InternVL3-7B) **(Ours)** | 73.2 | 65.0 | **68.8** | 91.8 | **77.6** | 64.5 | 71.8 |
| B3++ (Qwen2-7B) **(Ours)** | **74.1** | **70.0** | 66.5 | 84.6 | 75.9 | **67.1** | **72.0** (+1.3) |

Figure 2: We compare B3 and `Random Batches` (Qwen2-2B, 2 epochs) across batch different sizes. The performance gap is highest at smaller batch sizes and remains significant even as batch size increases. At a batch size of 64, B3 surpasses the 2B state-of-the-art, LLaVE (Llava-OV-2B).

These results demonstrate the effectiveness of our approach across model sizes. The performance improvements are consistent across both in-domain and out-of-domain tasks. B3++ performs exceptionally well on retrieval, which is one of the most important applications of embedding models. The other VLM-based approaches evaluated rely primarily on either synthetic data generation or modeling innovations. In contrast, our method, B3, is a batch mining strategy and is therefore complementary to these techniques, making it amenable to integration with them.

## 4.2 Effect of Batch Size $|B|$

The core strength of B3 lies in its effective batch mining strategy. To assess the quality of the mined batches, we compare the performance of B3 and `Random Batches` across a range of batch sizes, starting from 32. The results, presented in Fig. 2, correspond to models trained for two epochs. At smaller batch sizes, B3 outperforms `Random Batches` by a substantial margin, achieving improvements of over 14 points. Even at a larger batch size of 1024, the gains remain notable,

Table 2: Dissecting our B3++ methodology. Adding instructions for positives (+2.7) and B3 batch mining (+2.5) result in the highest gains. Results with `Qwen2-2B`, $|B| = 1024$. B3 beats `Random Batches` (w/ 5 hard negatives from $S$) despite using half the compute for training.

| Model | Ret. | Cla. | VQA | Gro. | ID | OOD | Avg. |
|---|---|---|---|---|---|---|---|
| *Backbone: Qwen2-2B;* | | *Batch Size ($\|B\|$): 1024* | | | | | |
| B3++ | 70.85 | 67.00 | 61.19 | 79.88 | 72.11 | 63.09 | **68.10** |
| B3 | 70.55 | 66.89 | 61.53 | 78.15 | 71.80 | 62.97 | 67.87 |
| Random Batches (w/ 5 hn. from $S$) | 68.63 | 66.99 | 61.03 | 78.33 | 71.52 | 61.67 | 67.14 |
| Random Batches | 66.33 | 66.47 | 59.08 | 75.30 | 69.59 | 60.05 | 65.35 |
| Random Batches (w/o Instruction 3.3) | 65.82 | 62.81 | 52.50 | 77.73 | 68.48 | 55.27 | 62.61 |

Table 3: Retrieval performance (*i2t*: image-to-text, *t2i*: text-to-image) on Flickr, COCO, and Urban1k datasets. B3++ (Qwen2-2B) surpasses current state-of-the-art models, while B3++ (Qwen2-7B) achieves the best overall results. Top scores are **bolded**, and second-best scores are underlined.

| Method | Flickr | | COCO | | Urban1k | |
|---|---|---|---|---|---|---|
| | t2i | i2t | t2i | i2t | t2i | i2t |
| CLIP(ViT-BigG/14) (2.5B) | 79.5 | 92.9 | 51.3 | 67.3 | 77.8 | 80.7 |
| UniME (Phi3.5-4B) | 77.0 | 88.2 | 49.8 | 66.8 | 92.7 | 95.1 |
| B3++ (Qwen2-2B) | 82.8 | 94.9 | 59 | 73.6 | 96.3 | 96.1 |
| EVA-CLIP [27] (8B) | 80.3 | 94.5 | 52.0 | 70.1 | 80.4 | 77.8 |
| UniME (Llava-Next-7B) | 81.9 | 93.4 | 53.7 | 70.1 | 95.2 | 95.9 |
| B3++ (Qwen2-7B) | **85.5** | **95.9** | **62.8** | **77.6** | **98.1** | **98.0** |

exceeding 2.5 points. **These results indicate that** B3 **consistently enhances performance across all batch sizes, with particularly pronounced benefits at smaller scales.**

For additional context, we include the performance of the current 2B state-of-the-art model, LLaVE, trained with its default configuration, as a horizontal reference line in the plot. **Remarkably,** B3 **surpasses** LLaVE**, the current 2B state-of-the-art model, even at a batch size as small as 64, further underscoring the effectiveness of its batch mining strategy**. Enabling effective training with smaller batches facilitates model development on limited hardware resources and allows scaling to substantially larger models in high-capacity environments.

### 4.3  Dissecting B3++

Table 2 shows the results for individual components of the proposed B3++ methodology. B3 exhibits performance only slightly below that of B3++. Both our models outperform `Random Batches` baselines. B3 consuming the exact same compute as `Random Batches` beats it by 2.5 points. B3 beats the `Random Batches` + (w/ 5 hn from $S$) which samples negatives from matrix $S$, by 0.7 points despite using half the compute. B3 **is both effective and efficient**.

### 4.4  Short and Long Caption Retrieval

Following `UniME` [6], we perform zero-shot evaluation of B3++ on short (Flickr [23], COCO [15]) and long (Urban1k [35]) image caption retrieval. As shown in Table 3, B3++ **(Qwen2-2B) surpasses** `UniME` **(7B), and** B3++ **(Qwen2-7B) achieves the best overall performance**. Consistent with the substantial retrieval gains shown in Table 1, B3++ outperforms all other baselines.

### 4.5  Ablations and Analysis

#### 4.5.1  Effect of $K$

As discussed in §3.1.2, excessively large values of $K$ may be suboptimal. In Table 1, we selected the value of $K$ using a held-out subset of the training data. We now provide empirical evidence to support this choice on the test set. As shown in the ablation results, increasing $K$ initially improves performance; however, beyond a certain threshold, **very high values of $K$ lead to a decline in performance**. More details on this in §F.

Table 4: 1.B3 batch selection performed with different sized clusters. Higher $K$ is better. Too large $K$ is also bad. Best $K$ lies in between. 2. Using a higher max resolution improves performance. 3. Our results indicate minimal performance difference between strong and weak teacher models.

| Ablation | Model | Ret. | Cla. | VQA | Gro. | ID | OOD | Avg. |
|---|---|---|---|---|---|---|---|---|
| 1 | B3 ($|B|$ = 512; $K$=8) | 69.81 | 66.94 | 59.93 | 80.00 | 71.01 | 62.89 | 67.40 |
|   | B3 ($|B|$ = 512; $K$=32) | 70.38 | 66.10 | 60.71 | 80.03 | 70.90 | 63.43 | **67.58** |
|   | B3 ($|B|$ = 512; $K$=512) | 69.74 | 66.39 | 59.70 | 81.03 | 70.60 | 63.13 | 67.28 |
| 2 | B3 (max resolution=700) | 69.96 | 66.60 | 60.78 | 78.70 | 71.30 | 62.64 | 67.45 |
|   | B3 (max resolution=1000) | 70.55 | 66.89 | 61.53 | 78.15 | 71.80 | 62.97 | **67.87** |

#### 4.5.2 Resolution

We examine the impact of limiting the maximum input resolution, using Qwen2-2B as the backbone. Images exceeding the specified resolution are down-sampled accordingly. Given a patch size of $28 \times 28$, Qwen2-2B produces approximately 600 tokens at a resolution of 700 and 1200 tokens at a resolution of 1000. Both experiments are conducted with a batch size of $|B| = 1024$. **Increasing the number of tokens per image consistently leads to improved performance**.

#### 4.5.3 Strength of Teacher

In our main results, we employed a teacher model of the same scale as the student—trained on the same dataset and without hard negatives i.e. VLM2Vec, to isolate the contribution of our batch mining approach. In this ablation, we replace the teacher with other weaker and stronger models. In B3, the teacher's sole role is to spot potential strong negatives and group them in the same batch. We simply take ranks from the teacher to build batches with strong in-batch negatives— and even with as few as 8 such negatives (in Table 4), performance remains solid. Critically, **the teacher's task is limited: it only needs to pick out some strong negatives, not all of them**.

When trained with a weaker teacher such as CLIP—whose performance on MMEB is markedly lower than B3 continues to deliver competitive outcomes in Table 5. These results indicate that B3's effectiveness is not contingent on a highly capable teacher.

Conversely, when trained with a stronger teacher, VLM2Vec(7B), the performance gains are minimal. We hypothesize that this is because the VLM2Vec(2B) teacher already provides sufficiently strong negatives, leaving little additional benefit from the larger VLM2Vec(7B) model.

Table 5: Performance of **B3 (2B)** with weaker CLIP(400M) and stronger VLM2Vec(7B) teachers. B3 even works with weaker teachers registering a strong performance. Stronger teachers don't make much difference.

| Model | Teacher | |B| | K | Ret. | Cla. | VQA | Gro. | ID | OOD | Avg. |
|---|---|---|---|---|---|---|---|---|---|---|
| Random Batches(2B) | - | 1024 | - | 66.33 | 66.47 | 59.08 | 75.30 | 69.59 | 60.05 | 65.35 |
| B3 (2B) | VLM2Vec(2B) | 512 | 8 | 69.81 | 66.94 | 59.93 | 80.00 | 71.01 | 62.89 | 67.40 |
| B3 (2B) | VLM2Vec(2B) | 512 | 32 | 70.38 | 66.10 | 60.71 | 80.03 | 70.90 | 63.43 | 67.58 |
| B3 (2B) | CLIP(400M) | 512 | 32 | 69.79 | 66.34 | 60.48 | 79.70 | 71.27 | 62.45 | 67.30 |
| B3 (2B) | VLM2Vec(7B) | 512 | 32 | 70.36 | 66.60 | 61.08 | 77.83 | 71.46 | 62.70 | 67.57 |

### 4.6 Comparison with other Batch Selection techniques, Hyperparameters

In this subsection, we compare B3 with alternative batch selection strategies and analyze the impact of its key hyper-parameters. For baseline comparison, we consider Grit Batch Mining (GBM), based on GritLM [22], which selects random targets from the same task as a batch selection mechanism. We also evaluate variants of B3 by varying its two primary hyper-parameters, $p$ and $m$. **As shown in Fig. 3, all configurations of B3 outperform GBM**. The results, averaged over 36 datasets, are statistically meaningful. Smaller values of $p$ can introduce false negatives, leading to a performance drop. This is evident in Fig. 3 especially at smaller batch sizes.

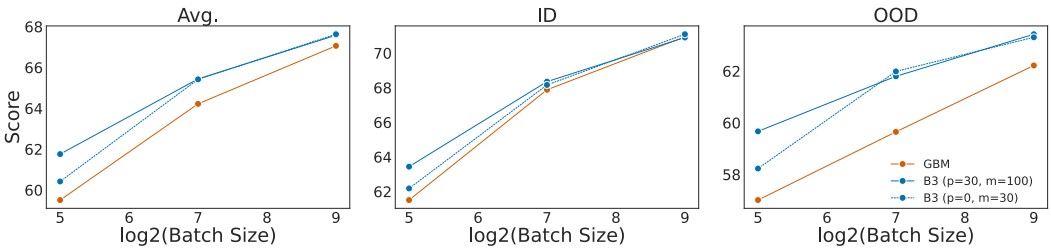

Figure 3: Variants of B3 hyper-parameters and the GBM baseline are evaluated across multiple batch sizes. B3 consistently outperforms GBM at all batch sizes. At smaller batch sizes, the impact of false negatives—introduced by lower values of $p$ in B3—is more apparent.

## 4.7  Generalizability to other domains?

We report the performance of B3, trained on the same NLI data as SimCSE (both with and without annotated hard negatives (HN)), evaluated on the STS benchmark. The corresponding SimCSE models (with/without HN) serve as the teacher models. All models (teacher and student) in the following table are 300M Roberta-Large. B3 demonstrates notable improvements over SimCSE on the STS benchmark, validating our batch mining approach for text-only tasks. Note that even starting with HN in the teacher, we can still see improvements. In future work, we plan to extend this to larger benchmarks like MMEB.

Table 6: Sentence similarity results on the STS benchmarks for B3 and SimCSE variants. $()_{HN}$ denotes that the model was trained using one hard negative from [5]. Batch size , $|B| = 512$ for all models and all used the 275K training dataset from [5]

| Model | Teacher | STS12 | STS13 | STS14 | STS15 | STS16 | STSB | SICKR | Avg. |
|---|---|---|---|---|---|---|---|---|---|
| SimCSE | - | 75.63 | 84.40 | 78.06 | 84.82 | 81.87 | 84.21 | 77.00 | 80.86 |
| B3 | SimCSE | 77.49 | 85.73 | 80.37 | 85.36 | 82.75 | 84.24 | 75.40 | 81.62 |
| $SimCSE_{HN}$ | - | 77.30 | 86.68 | 82.20 | 86.58 | 84.02 | 86.36 | 81.84 | 83.57 |
| $B3_{HN}$ | $SimCSE_{HN}$ | 78.70 | 87.58 | 83.47 | 87.22 | 84.50 | 86.85 | 81.33 | 84.24 |

## 4.8  Discussion - Runtime, Implementation and Future work.

The entire B3 methodology operates as an offline preprocessing step over the training dataset. The teacher model's scoring of all training examples is fully parallelizable, and generating the rank list requires only $\mathcal{O}(n \log n)$ time. Subsequently, applying the METIS algorithm on the sparse graph $S$ incurs a linear $\mathcal{O}(n)$ runtime. The resulting mined batches can be efficiently stored on disk in a structured format. Integrating B3 into contrastive training pipelines requires minimal modifications—training simply involves sampling from the preprocessed batches. Looking ahead, we aim to extend the B3 framework to both text-only scenarios and broader multimodal settings.

## 5  Conclusion

We propose B3, an effective batch mining technique that leverages the entire training dataset to form batches composed of mutually strong negatives, to improve contrastive learning. B3++, a variant of B3 using hard-negatives achieves state-of-the-art results on the MMEB embedding benchmark comprising 36 diverse embedding tasks. B3 is complementary with existing methods centered on data synthesis and model architecture, and can be seamlessly integrated on top of them to further enhance performance. Through extensive experiments, we demonstrate that B3 consistently outperforms existing batch mining strategies across a wide range of batch sizes. Notably, B3++ surpasses the current 2B state-of-the-art model even with a batch size of just 64. Furthermore, B3 outperforms a random batch baseline augmented with five hard negatives, despite not using any hard negatives and requiring only half the training time. B3++ also outperforms other methods on image caption retrieval datasets, further showcasing its versatility and strength.

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

## A   Code

Code: https://github.com/raghavlite/B3

## B   Limitations

The major limitation of B3 is the extra time and compute required to perform ranking on the entire training set. However, as discussed in §4.8, this could be parallelized and achieved in almost $\mathcal{O}(n \log n)$ time.

## C   Model Details

We test our methodology using three different VLMs - `Qwen2-VL-7B-Instruct`, `Qwen2.5-7B-Instruct`, `InternVL3-8B` with both 7B and 2B variants wherever available. We use $m = 100$ following SFR-Embedding and NV-Embed. For $p$, we tuned this value on heldout portions of trainset. We used $p = 30$ for retrieval and grounding tasks and $p = 70$ for VQA tasks. For classification tasks, we just filter out the golden label from the rank list. All models in this work are trained using LoRA with a rank of 8. Unless mentioned otherwise, models are trained for 2k steps with peak learning rate of 1e-4 and warmup of 10%. Temperature used was 0.02. The last layer hidden state of the last token is the representation that is tuned in contrastive training. For the qwen models, we use the default maximum of 1.2k tokens (max resolution is 1000) to represent an image. For InternVL models, we use the default max pixels of 3k as recommended. Due to the heavy compute required for InternVL models, we only evaluate them with B3 and not B3++.

## D   Compute and Runtime Details

All training and evaluation were conducted on 8 H200 GPUs. B3++ was trained for 24 hours using Qwen-2B and 40 hours using InternVL-2B. In comparison, B3 (without hard negatives) required approximately half the training time of B3++ for the same backbone. The 7B variants of each method took roughly twice as long to train as their corresponding 2B counterparts.

# E  Proof of Theorem 1

We now derive an upper bound for Eq. 2. Let $H_i^K$ and $H_{B_i,i}^K$ denote the sum of the top $K$ exponent terms in the denominator for the global and batch loss components, respectively, for each query $x_i$. The sum of other terms is $L_i^K$ and $L_{B_i,i}^K$ for the same respectively.

$$\mathcal{L}^{Global} - \mathcal{L}^B = \sum_{i=1}^{N} -\log \frac{exp(x_i^T y_i)}{\sum_{j=1}^{N} exp(x_i^T y_j)} - \sum_{i=1}^{N} -\log \frac{exp(x_i^T y_i)}{\sum_{j \in B_i} exp(x_i^T y_j)} \tag{7}$$

$$= \sum_{i=1}^{N} \left( \log \left( \sum_{j=1}^{N} exp(x_i^T y_j) \right) - \log \left( \sum_{j \in B_i} exp(x_i^T y_j) \right) \right) \tag{8}$$

$$= \sum_{i=1}^{N} \log \left( H_i^K + L_i^K \right) - \log \left( H_{B_i,i}^K + L_{B_i,i}^K \right) \tag{9}$$

$$\leq \sum_{i=1}^{N} \log \left( \frac{N}{K} H_i^K \right) - \log \left( H_{B_i,i}^K + L_{B_i,i}^K \right) \tag{10}$$

$$= \sum_{i=1}^{N} \log \left( \frac{N}{K} \frac{H_i^K}{H_{Bi}^K} \right) - \log \left( 1 + \frac{L_{B_i,i}^K}{H_{B_i,i}^K} \right) \tag{11}$$

$$\leq \sum_{i=1}^{N} \log \left( \frac{N}{K} \frac{H_i^K}{H_{B_i,i}^K} \right) \tag{12}$$

Note that this bound holds for all $K$ is a fairly tight bound given the peaked distributions of contrastive trained models (i.e. $\log \left( 1 + \frac{L_{B_i,i}^K}{H_{B_i,i}^K} \right) \approx 0$) shown in §E.1. Although the plot in §E.1 correspond to ones from a trained model, such peaked distribution is achieved fairly quickly during the initial rounds of training.

## E.1  Peaked Distributions

Fig. 4 and Fig. 5 show the heavily peaked score distributions of similarity a query ($x_i$) against all positives in the dataset ($y_i, \forall i$) contrastive training datasets. Exponent of this similarity score will further make the distribution peaked and $\frac{L_{B_i,i}^K}{H_{B_i,i}^K} \to 0$

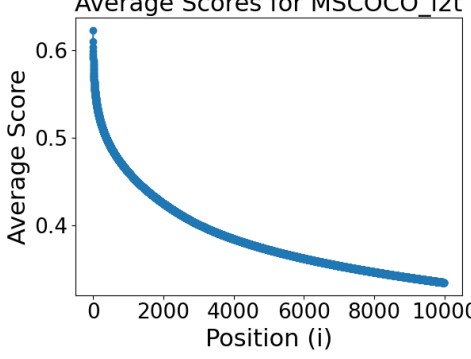

Figure 4: Average top-10000 scores for MSCOCO_i2t with VLM2Vec-Qwen2B

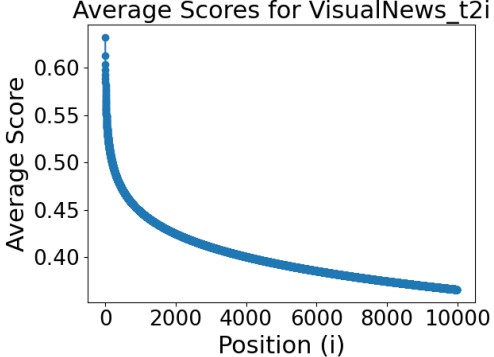

Figure 5: Average top-10000 scores for Visual-News_t2i with VLM2Vec-Qwen2B

# F  Choice of K (More Details)

$$\mathcal{L}_{\text{sup}}(\hat{f}) \leq \alpha L_{\text{un}}^{\neq}(f) + \beta s(f) + \eta \operatorname{Gen}_M \tag{13}$$

where $\mathcal{L}_{\text{sup}}(\hat{f})$ is a measure of downstream performance. While other terms are less relevant to our work, the term $\beta s(f)$ is strongly affected by $K$. $\beta$ increases with the number of hard negatives ($K$ in our case). $s(f)$ is the a measure of intra-class covariance. A higher value of $K$ would result in the query contrasting against large number of negatives causing a high value of intra-class covariance (similar examples fall into the same latent class). Overall, Saunshi et al. [26] suggests that $K$ cannot be very big. This observation is also empirically corroborated by SFR-Embedding [19], where using high number of hard negatives resulted in inferior performance. Owing to this constraint, we empirically tune $K$.

# G  Effect of $p$ and $m$ on Model Performance

We conduct a detailed analysis of the hyperparameters $p$ and $m$, which control the sampling of positives and the exclusion of potential false negatives in B3. This mechanism, also adopted in prior contrastive methods such as **SFR-Embed** and **NV-Embed**, helps mitigate false-negative contamination within a batch.

## G.1  Why VQA Tasks Benefit from Larger $p$

For retrieval tasks, we adopt the same value of $p$ as used in SFR-Embed, where smaller values (e.g., $p \in [30, 70]$) were shown to perform well. However, Visual Question Answering (VQA) datasets within MMEB exhibit a distinct property: the positive (target) responses are typically short, often single words. Table 7 summarizes the average target lengths across VQA datasets, compared to those in image-to-text retrieval.

Table 7: Average target lengths in VQA datasets.

| Dataset | OK-VQA | A-OKVQA | DocVQA | InfoVQA | ChartQA | Visual7W | **Avg.** |
|---------|--------|---------|--------|---------|---------|----------|----------|
| **Length** | 1.24 | 1.28 | 2.16 | 1.61 | 1.12 | 1.98 | 1.57 |

Table 8: Average target lengths in image-to-text retrieval datasets.

| Dataset | VisualNews_i2t | MSCOCO_i2t | WebQA | **Avg.** |
|---------|----------------|------------|-------|----------|
| **Length** | 18.89 | 10.44 | 23.45 | 17.60 |

Because VQA answers are extremely short, many targets across examples share nearly identical surface forms or meanings, e.g., *"ponytail," "pony tail," "ponytail"*. This increases the likelihood of false negatives when contrasting across the batch. Consequently, a higher value of $p$ is beneficial for VQA tasks, as it reduces the probability of penalizing semantically equivalent positives.

For computational efficiency, we tuned $p$ using smaller batch sizes and reused these settings for larger batches. Table 9 shows results from models trained and evaluated solely on VQA datasets at batch size $|B| = 32$.

Table 9: Effect of $p$ on VQA performance at smaller batch size.

| $p$ | $|B|$ | **ID** | **OOD** | **Avg.** |
|-----|-------|--------|---------|----------|
| 30 | 32 | 57.80 | 56.73 | 57.37 |
| 70 | 32 | 58.38 | 57.95 | 58.21 |

Performance improves with larger $p$ at smaller batch sizes. However, this sensitivity diminishes as $|B|$ increases, as discussed next.

## G.2 Joint Analysis of $p$ and $m$

We next analyze the joint influence of $p$ and $m$ on retrieval datasets. Table 10 shows results for models trained and tested with $|B| = 32$.

Table 10: Effect of $m$ on retrieval performance at small batch size.

| $p$ | $m$ | $|B|$ | ID | OOD | Avg. |
|---|---|---|---|---|---|
| 30 | 100 | 32 | 72.83 | 55.50 | 67.05 |
| 30 | 500 | 32 | 72.25 | 54.65 | 66.38 |
| 30 | 800 | 32 | 72.60 | 55.15 | 66.78 |

At smaller batch sizes, performance remains sensitive to $m$, consistent with SFR-Embed where $m = 100$ was found effective. These settings continue to yield robust results for B3.

## G.3 Behavior When $p = 0$

To understand the effect of disabling the exclusion mechanism, we evaluate performance with $p = 0$ across multiple batch sizes. The results in Table 11 show that increasing $p$ is beneficial for small batches but has little influence at larger ones.

Table 11: Effect of $p$ and $p + m$ across different batch sizes.

| $p_{(Ret,VQA)}$ | $p+m_{(Ret,VQA)}$ | $|B|$ | ID | OOD | Avg. |
|---|---|---|---|---|---|
| 0,0 | 30,70 | 32 | 62.18 | 58.23 | 60.42 |
| 30,70 | 130,170 | 32 | 63.44 | 59.67 | 61.76 |
| 0,0 | 30,70 | 128 | 68.16 | 61.99 | 65.42 |
| 30,70 | 130,170 | 128 | 68.35 | 61.80 | 65.44 |
| 0,0 | 30,70 | 512 | 71.09 | 63.31 | 67.63 |
| 30,70 | 130,170 | 512 | 70.90 | 63.43 | 67.58 |

At higher batch sizes ($|B| = 512$), the model becomes highly robust to variations in $p$ and $m$, as shown in Table 12.

Table 12: Performance stability of B3 at large batch size ($|B| = 512$).

| $p_{(Ret,VQA)}$ | $p+m_{(Ret,VQA)}$ | $|B|$ | ID | OOD | Avg. |
|---|---|---|---|---|---|
| 0,0 | 30,70 | 512 | 71.09 | 63.31 | 67.63 |
| 0,0 | 100,100 | 512 | 71.06 | 63.24 | 67.59 |
| 30,70 | 130,170 | 512 | 70.90 | 63.43 | 67.58 |

## G.4 Summary

The above analysis reveals that $p$ and $m$ exert a stronger influence at smaller batch sizes, where proper tuning helps reduce false negatives and improve representation alignment. At larger batch sizes, B3 exhibits remarkable robustness to these hyperparameters, achieving consistent performance even without fine-tuning. This highlights B3's scalability and stability under large-batch contrastive learning regimes.

# H Choice of the Clustering Algorithm

Note that we need the cluster size $K$ to be fixed in B3 (as the batch size $|B|$ is fixed). Most clustering algorithms, such as K-Means or Agglomerative, have balanced variants, but these are typically much more computationally expensive, often running in $O(n^3)$. In contrast, METIS operates in $O(n)$ for sparse graphs while producing equal-sized clusters. A key idea within B3 cluster construction is to minimize the presence of false negatives (upto rank $p$ ) in the training clusters. Note that each

example in the training dataset has a different set of false negatives. Clustering algorithms like K-Means/Agglomerative would require substantial modifications to achieve this. Examples with ranks beyond $m$ in B3 are less relevant and should not play a role in clustering. Here again, it is non-trivial to include these constraints in regular clustering algorithms. To accommodate rank-based constraints, we opted for graph-based clustering algorithms. Furthermore, to satisfy the fixed batch size requirement, we selected the METIS algorithm.

## I  Prompts (More Details)

In most tasks where the targets comprised either images or image-text pairs, VLM2Vec [8] employed explicit instruction prompts. However, for tasks involving purely text-based positives, no such prompts were used. Table 13 enumerates the tasks along with the corresponding prompts used in our experiments. The majority of these tasks fall under classification and visual question answering.

| Category | Dataset | Instruction Prompt |
|---|---|---|
| Retrieval | MSCOCO_i2t
VisualNews_i2t | Represent the image caption: |
| Classification | ImageNet-1K
HatefulMemes
SUN397
N24News
VOC2007
Place365
ImageNet-A
ImageNet-R
ObjectNet
Country211 | Represent the class label: |
| Visual Question Answering | OK-VQA
A-OKVQA
DocVQA
InfographicsVQA
ChartQA
Visual7W
ScienceQA
GQA
TextVQA
VizWiz | Represent the answer: |

Table 13: Additional positive instruction prompts that were used in B3, B3++. For positives of other datasets, we used the existing prompts from VLM2Vec [8]. These prompts are expected to decouple diverse tasks during training.

## Broader Impact and Discussion of Ethics

While our model is not tied to any specific applications, it could be used in sensitive contexts such as health-care, etc. Any work using our method is requested to undertake extensive quality-assurance and robustness testing before applying in their setting. To the best of our knowledge, the datasets used in our work do not contain any sensitive information.

