# OpenReview forum: "Breaking the Batch Barrier (B3) of Contrastive Learning via Smart Batch Mining"
_NeurIPS.cc/2025/Conference — NeurIPS 2025 spotlight_

### Official Review · Reviewer_4DdK · 2025-06-30

**Clarity:** 2
**Significance:** 3
**Originality:** 3
**Rating:** 5
**Confidence:** 2

**Summary:**

This paper presents "Breaking the Batch Barrier" (B3), a batch construction strategy for contrastive learning. The B3 method employs a pretrained teacher model to create a similarity ranking for all examples in the dataset. This ranking is used to construct a sparse similarity graph, which is then used for clustering examples that are strong mutual negatives using a community detection algorithm (METIS). Training batches are then formed by sampling from these communities, achieving challenging negatives without the computational overhead of adding individually mined hard negatives. The authors also propose B3++, an optional extension that incorporates a unified hard-negative mining strategy for further performance gains. The B3 is evaluated on the MMEB multimodal benchmark, and it demonstrates significant improvements over existing methods.

**Questions:**

Q1: The negative samples exclusion ranking threshold p needs to be tuned differently for various task types (e.g., p=30 for retrieval, p=70 for VQA). This suggests the method is sensitive to the data's domain and structure. Could you provide more insight into the underlying reasons?

**Ethical Concerns:**

["NO or VERY MINOR ethics concerns only"]

**Final Justification:**

The author has provide more context on the design of of excluding top p ranks in the negative sampling, and performed additional experiments to show that  p=0 works very well at larger batch sizes.

**Limitations:**

yes

**Paper Formatting Concerns:**

None.

**Quality:**

3

**Strengths And Weaknesses:**

Strength

S1: The proposed method leverages graph-based community clustering to construct batches that are rich in hard negative, which is efficient compared to existing hard-negative samples mining methods.

S2: The author performed extensive experiments to show that the B3 method achieves sota performance despite training at smaller batch size.

S3: The ablation studies and comparisons across various batch sizes and hyperparameters validate the effectiveness of B3.

Weakness

W1:  I was very confused about the positive samples at the beginning. The papers uses the standard terminology "(query, positive) pairs" to refer to a "positive pair", but does not explicitly explain that a "positive" can be of a different data type than its corresponding "query" (e.g., an image paired with a text caption). For readers whose main exposure to contrastive learning comes from self-supervised methods where positive pairs are typically of the same modality, this lack of clarity may lead to a fundamental confusion.

W2: The design of excluding top p ranks from the negative samples is quite ad-hoc, and has to be tuned case by case.

---

> ### Author Rebuttal · Authors · 2025-07-30
>
> Thank you for your thoughtful comments and time reviewing our work. We are truly pleased with the positive reception our work has garnered from you and the other reviewers.
> * “method gives new SOTA performance…method is simple…should yield richer gradients…” - RM3oh
> * “B3 method is rigorously formulated…supported with theoretical bounds on InfoNCE loss approximation…consistent and substantial gains, especially under small batch sizes…graph-based community detection is a novel framing ” - RipHG
> * “has clear intuition and solid theoretical justification…Experiments seem solid and extensive…” - Rm62i
> * “achieves sota performance despite training at smaller batch size…ablation studies validate the effectiveness of B3” - R4DdK
>
> ---
> ---
>
> >W1: ... confused about the positive samples at the beginning. The papers uses the standard terminology "(query, positive) pairs" to refer to a "positive pair", but does not explicitly explain that a "positive" can be of a different data type than its corresponding "query" (e.g., an image paired with a text caption). Readers whose main exposure to contrastive learning comes from self-supervised methods... this lack of clarity may lead to a fundamental confusion.
>
> Thank you for highlighting this point. As the reviewer correctly observed, the query and positive can originate from different modalities, and each may also be a combination of modalities. For example, in VQA, the query comprises an image and a text question, while the positive is a text answer. This is an important characteristic of the MMEB training set, which we will clarify in the final draft.
>
> ---
> ---
>
> > The design of excluding top p ranks from the negative samples is quite ad-hoc, and has to be tuned case by case. Q1: The negative samples exclusion ranking threshold p needs to be tuned differently for various task types (e.g., p=30 for retrieval, p=70 for VQA). This suggests the method is sensitive to the data's domain and structure. Could you provide more insight into the underlying reasons?
>
> Excluding top $p$ ranks serves as a mechanism for excluding false negatives, **a strategy previously employed in SFR‑Embed and NV‑Embed**. For retrieval, we adopted the same value of $p=30$ as used in SFR‑Embed. NV‑Embed also showed that lower values of $p$ between $10$ and $30$ perform well. However, in the case of VQA training datasets in MMEB, the positive (target) responses were notably short. The table below reports the average target lengths for the VQA tasks.
>
> | Dataset         | OK-VQA | A-OKVQA | DocVQA | InfoVQA | ChartQA | Visual7W | Avg.  |
> |:------------|:-----:|:------:|:----:|:--------:|:---:|:--------:|:-------:|
> | Length|1.24| 1.28| 2.16   | 1.61| 1.12|1.98| 1.57 |
> |||||||||
>
> In contrast, for image2text retrieval tasks, the average positive (target) length was as follows
> | Dataset| VisualNews_i2t | MSCOCO_i2t | WebQA | Avg.       |
> |:------------|:---------:|:-----:|:----:|:------:|
> | Length| 18.89 | 10.44| 23.45 | 17.6 |
> ||||||
>
> Due to this reason, **we found a significantly higher number of targets which had similar/same meaning as that of the labelled positive**. For instance, here is the first example in OK-VQA dataset. As seen, due to the short nature of the answers a higher number of targets/positives of other examples are very similar/same compared to retrieval.
>
> - Image: Image containing a women playing tennis
> - Q: What is the hairstyle of the blond called?
> - Top ranked targets/positives: [“ponytail”, “ponytail”, “ponytail”, “pony tail”, “pony tail”,....]
>
> Hence, we used a higher value of $p$ for VQA tasks. For computational efficiency, we tuned $p$ for smaller batch sizes and retained these values for larger batch sizes. Here are results for a model trained and tested on VQA datasets only at batch size ($|B|=32$) for different $p$ values.
>
> | p   | \|B\|  | ID    | OOD   | Avg.  |
> |:-----|:-----:|:----:|:-------:|:-------:|
> | 30  | 32  | 57.80 | 56.73 | 57.37 |
> | 70  | 32  | 58.38 | 57.95 | 58.21 |
> ||||||
>
> At lower batch sizes, results improve for higher $p$ values. However, as discussed next, at higher batch sizes, $p$ has a smaller effect on the performance.
>
> The table below reports results for $p=0$ across different batch sizes. At smaller batch sizes, using an elevated value of $p$ proves beneficial, effectively reducing multiple false negatives. However, at larger batch sizes, this effect diminishes; as shown, **performance remains comparable between $p=0$ and $p=30$ when using higher batch sizes**.
>
> | p (Ret,VQA)   | p+m (Ret, VQA)   | \|B\|  |   ID   |   OOD  |  Avg.  |
> |:-----|:-----:|:-----:|:--------:|:--------:|:--------:|
> | 0,0   | 30,70  | 32   | 62.18  | 58.23  | 60.42  |
> | 30,70  | 130,170 | 32  | 63.44  | 59.67  | 61.76  |
> | 0,0   | 30,70  | 128  | 68.16  | 61.99  | 65.42  |
> | 30,70  | 130,170 | 128 | 68.35  | 61.80  | 65.44  |
> | 0,0   | 30,70  | 512 | 71.09 | 63.31 | 67.63 |
> | 30,70   | 130,170 | 512 | 70.90 | 63.43 | 67.58 |
> |||||||
>
> Presented below are additional variations of $p$ and $m$ at a higher batch size (512). As shown, **B3 maintains consistent performance across these variations when using a larger batch size.**
>
> | p (Ret, VQA)   | p+m (Ret, VQA)  | \|B\|  | ID    | OOD  | Avg.  |
> |:-----|:-----:|:-----:|:-------:|:-------:|:-------:|
> | 0,0   | 30,70  | 512 | 71.09 | 63.31 | 67.63 |
> | 0,0   | 100,100 | 512 | 71.06 | 63.24 | 67.59 |
> | 30,70  | 130,170 | 512 | 70.90 | 63.43 | 67.58 |
> |||||||
>
> As shown p=0 works very well at larger batch sizes.
>
> **Main Takeaway (from $p$ and $m$ analysis)**
> - *The previous tables indicate that the hyperparameters $p$ and $m$ have a greater impact on performance at smaller batch sizes. While B3 requires tuning of these parameters to achieve optimal results at smaller batch sizes, it shows increased robustness to their variation at larger batch sizes.*
>
> ---
> ---
> To further strengthen our B3 pitch, we have included two additional results as requested by other reviewers.
>
> > Generalizability to other domains?
>
> We believe that B3 is a highly generalizable method for contrastive training. To further substantiate our claim, we report the performance of B3, trained on the same NLI data as SimCSE (both with and without annotated hard negatives (HN)), evaluated on the STS benchmark. The corresponding SimCSE models (with/without HN) serve as the teacher models. All models (teacher and student) in the following table are 300M Roberta-Large.
>
>
> | Model   |  |  Teacher | | Data  | \|B\| |  | |STS12 | STS13 | STS14 | STS15 | STS16 | STSB | SICKR | Avg.   |
> |:--|:--:|:---|--|:---:|:--:|:--:|:--:|:---:|:---:|:--:|:--:|:---:|:----:|:-:|:---:|
> | SimCSE (w/o HN) | | - | | 275k | 512  | | |75.63 | 84.40 | 78.06 | 84.82 | 81.87 | 84.21 | 77.00 | 80.86  |
> | **B3 (w/o HN)** | | SimCSE (w/o HN) | | 275k | 512 | | | 77.49 | 85.73 | 80.37 | 85.36 | 82.75 | 84.24  | 75.40 | **81.62** |
> ||||||||||||||||
> | SimCSE | | - | | 275k | 512  | | | 77.30 | 86.68 | 82.20 | 86.58 | 84.02 | 86.36 | 81.84  | 83.57  |
> | **B3** | | SimCSE  | | 275k | 512  | | | 78.70 | 87.58 | 83.47 | 87.22 | 84.50 | 86.85 | 81.33  | **84.24** |
> ||||||||||||||||
>
>
> **B3 demonstrates notable improvements over SimCSE on the STS benchmark**, validating our batch mining approach for text-only tasks. Note that even starting with HN in the teacher, we can still see improvements. In future work, we plan to extend this to larger benchmarks like MMEB.
>
> ---
> ---
>
> > Need for a strong teacher?
>
> In B3, the teacher’s sole role is to spot potential strong negatives and group them in the same batch. We simply take ranks $p:(p+m)$ from the teacher to build batches with strong in-batch negatives— and even with as few as 8 such negatives ($K=8$ in the Table), performance remains solid. Critically, the **teacher’s task is limited: it only needs to pick out some strong negatives, not all of them**.
>
> Hence, this **doesn’t require a very accurate teacher**, as the reviewer suspected. To prove the point, we run B3 with a much weaker teacher (CLIP). Despite CLIP performing far below VLM2Vec on MMEB, using it as a teacher still delivers strong results.
>
>
> | Model           |      | Teacher |  | \|B\|  | | K  |  |  | **Ret.** | **Cla.** | **VQA** | **Gro.** | **ID** | **OOD** | **Avg** |
> |:--|--|:---:|---|:--:|:---:|:--:|:---:|:--:|:---:|:---:|:----:|:---:|:---:|:---:|:--:|
> | Random Batches (2B) | | - | | 1024 | | - | | |66.33 | 66.47|59.08|75.30|69.59|60.05|65.35|
> | B3 (2B) | | VLM2Vec2B | | 512 | | 8| | |69.81|66.94|59.93|80.00|71.01|62.89|67.40|
> | B3 (2B) | | VLM2Vec2B | | 512 | | 32  | | | 70.38  | 66.10 | 60.71 | 80.03 | 70.90 | 63.43 | 67.58  |
> | B3 (2B) | | VLM2Vec7B | | 512 | | 32 | | | 70.36 | 66.60  | 61.08 | 77.83  | 71.46 | 62.70 | 67.57 |
> | B3 (2B) | | **CLIP400M** | | 512 | | 32 | | | 69.79|66.34| 60.48 | 79.70 | 71.27 | 62.45 | **67.30** |
> ||||||||||
>
>
> All that said, a strong teacher model can still be trained using only positives and a sufficiently large batch size, without relying on any negatives. VLM2Vec teacher models were trained solely with positives and random batches, without incorporating hard negatives.
>
> ---
> ---
>
> We again thank the reviewer for their time and effort. We will incorporate the additional results, analyses, and detailed responses provided here into the final draft to ensure a more comprehensive and rigorous presentation.

---

> > ### Comment · Reviewer_4DdK · 2025-08-06
> >
> > Thanks for the authors' patient and detailed response. My concerns regarding the design and impact of excluding top p ranks in the negative sampling is much dismissed and I will update my final rating accordingly.

---

> > > ### Author Response · Authors · 2025-08-06
> > > **Thank You**
> > >
> > > We sincerely thank the reviewer for acknowledging our clarifications on excluding top‑$p$ ranks in negative sampling. Your thoughtful feedback has greatly strengthened our draft.

---

> ### Comment · Area_Chair_WD3N · 2025-08-05
>
> Reviewer 4DdK, please engage in the discussion period. I understand that the review period started over a weekend, but we only have a few days remaining in the (now slightly extended) discussion period. The authors have provided a thoughtful response to your review and you are obligated to respond to it. You should share with the authors if they addressed questions or concerns you had, and seek clarification about any questions or concerns that remain. Please post your response as soon as you can so that there is time for the authors to follow up and discussion to progress as needed.

---

### Official Review · Reviewer_m62i · 2025-07-02

**Clarity:** 3
**Significance:** 3
**Originality:** 3
**Rating:** 5
**Confidence:** 2

**Summary:**

The paper introduces Breaking the Batch Barrier (B3), which uses a pretrained embedding model to construct a sparse similarity graph over the dataset and then applies graph clustering to form batches of strong negatives, thereby obtaining high-quality batches in a computationally efficient manner. Experiments show that B3 and its enhanced variant, B3++, outperform prior state-of-the-art methods across model scales and batch sizes while reducing overall computation.

**Questions:**

I wonder what the minimal requirements are for the teacher embedding model. How well does the method perform when the teacher is weaker or smaller than the model being trained—a scenario I think is very practically relevant.

**Ethical Concerns:**

["NO or VERY MINOR ethics concerns only"]

**Final Justification:**

I think this paper makes a novel contribution to the field and I am also satisfied with the rebuttal.

**Limitations:**

The authors have discussed the limitations in Appendix B.

**Quality:**

3

**Strengths And Weaknesses:**

Strengths

1. I find the  method well motivated, has clear intuition and solid theoretical justification.

2. Experiments seem solid and extensive to me. They cover multiple scales and batch sizes and demonstrate consistent improvements.

3. Ablation studies confirm the contribution of each component, reinforcing the method’s design choices.

Weaknesses

I don’t see any major weaknesses; however, I’m an expert this specific topic or familiar with all existing methods, so I’ll leave it to other reviewers to judge whether the comparisons are comprehensive and the experimental setup is adequate.

---

> ### Author Rebuttal · Authors · 2025-07-30
>
> Thank you for your thoughtful comments and time reviewing our work. We are truly pleased with the positive reception our work has garnered from you and the other reviewers.
> * “method gives new SOTA performance…method is simple…should yield richer gradients…” - RM3oh
> * “B3 method is rigorously formulated…supported with theoretical bounds on InfoNCE loss approximation…consistent and substantial gains, especially under small batch sizes…graph-based community detection is a novel framing ” - RipHG
> * “has clear intuition and solid theoretical justification…Experiments seem solid and extensive…” - Rm62i
> * “achieves sota performance despite training at smaller batch size…ablation studies validate the effectiveness of B3” - R4DdK
>
> To further strengthen our B3 pitch, we have included two additional results as requested by other reviewers.
>
> ---
> ---
>
> > Generalizability to other domains?
>
> We believe that B3 is a highly generalizable method for contrastive training. To further substantiate our claim, we report the performance of B3, trained on the same NLI data as SimCSE (both with and without annotated hard negatives (HN)), evaluated on the STS benchmark. The corresponding SimCSE models (with/without HN) serve as the teacher models. All models (teacher and student) in the following table are 300M Roberta-Large.
>
>
> | Model   |  |  Teacher | | Data  | \|B\| |  | |STS12 | STS13 | STS14 | STS15 | STS16 | STSB | SICKR | Avg.   |
> |:--|:--:|:---|--|:---:|:--:|:--:|:--:|:---:|:---:|:--:|:--:|:---:|:----:|:-:|:---:|
> | SimCSE (w/o HN) | | - | | 275k | 512  | | |75.63 | 84.40 | 78.06 | 84.82 | 81.87 | 84.21 | 77.00 | 80.86  |
> | **B3 (w/o HN)** | | SimCSE (w/o HN) | | 275k | 512 | | | 77.49 | 85.73 | 80.37 | 85.36 | 82.75 | 84.24  | 75.40 | **81.62** |
> ||||||||||||||||
> | SimCSE | | - | | 275k | 512  | | | 77.30 | 86.68 | 82.20 | 86.58 | 84.02 | 86.36 | 81.84  | 83.57  |
> | **B3** | | SimCSE  | | 275k | 512  | | | 78.70 | 87.58 | 83.47 | 87.22 | 84.50 | 86.85 | 81.33  | **84.24** |
> ||||||||||||||||
>
>
> **B3 demonstrates notable improvements over SimCSE on the STS benchmark**, validating our batch mining approach for text-only tasks. Note that even starting with HN in the teacher, we can still see improvements. In future work, we plan to extend this to larger benchmarks like MMEB.
>
> ---
> ---
>
> > Need for a strong teacher?
>
> In B3, the teacher’s sole role is to spot potential strong negatives and group them in the same batch. We simply take ranks $p:(p+m)$ from the teacher to build batches with strong in-batch negatives— and even with as few as 8 such negatives ($K=8$ in the Table), performance remains solid. Critically, the **teacher’s task is limited: it only needs to pick out some strong negatives, not all of them**.
>
> Hence, this **doesn’t require a very accurate teacher**, as the reviewer suspected. To prove the point, we run B3 with a much weaker teacher (CLIP). Despite CLIP performing far below VLM2Vec on MMEB, using it as a teacher still delivers strong results.
>
>
> | Model           |      | Teacher |  | \|B\|  | | K  |  |  | **Ret.** | **Cla.** | **VQA** | **Gro.** | **ID** | **OOD** | **Avg** |
> |:--|--|:---:|---|:--:|:---:|:--:|:---:|:--:|:---:|:---:|:----:|:---:|:---:|:---:|:--:|
> | Random Batches (2B) | | - | | 1024 | | - | | |66.33 | 66.47|59.08|75.30|69.59|60.05|65.35|
> | B3 (2B) | | VLM2Vec2B | | 512 | | 8| | |69.81|66.94|59.93|80.00|71.01|62.89|67.40|
> | B3 (2B) | | VLM2Vec2B | | 512 | | 32  | | | 70.38  | 66.10 | 60.71 | 80.03 | 70.90 | 63.43 | 67.58  |
> | B3 (2B) | | VLM2Vec7B | | 512 | | 32 | | | 70.36 | 66.60  | 61.08 | 77.83  | 71.46 | 62.70 | 67.57 |
> | B3 (2B) | | **CLIP400M** | | 512 | | 32 | | | 69.79|66.34| 60.48 | 79.70 | 71.27 | 62.45 | **67.30** |
> ||||||||||
>
>
> All that said, a strong teacher model can still be trained using only positives and a sufficiently large batch size, without relying on any negatives. VLM2Vec teacher models were trained solely with positives and random batches, without incorporating hard negatives.
>
> ---
> ---
> We again thank the reviewer for their time and effort. We will incorporate the additional results, analyses, and detailed responses provided here into the final draft to ensure a more comprehensive and rigorous presentation.

---

### Official Review · Reviewer_ipHG · 2025-07-05

**Clarity:** 2
**Significance:** 2
**Originality:** 2
**Rating:** 4
**Confidence:** 4

**Summary:**

This paper introduces B3, a novel batch mining strategy for contrastive learning (CL), particularly in multimodal settings. Traditional CL relies heavily on in-batch negatives, where performance scales with batch size. B3 circumvents this by constructing semantically rich batches via a graph-based approach. It includes several steps: ranking training samples via a pretrained teacher model, constructing a sparse graph, identifying strong negatives through METIS clustering and forming sampled batches. The method, particularly its enhanced variant B3++, achieves state-of-the-art performance on the MMEB multimodal benchmark, often outperforming existing methods while requiring significantly smaller batch sizes (e.g., 64 vs. 1024). The paper supports its claims with theoretical justification, ablation studies, and evaluation across multiple model scales.

**Questions:**

1.	Can B3 adapt to dynamic or evolving datasets, such as in online or streaming learning scenarios? B3 relies on full-dataset ranking and offline clustering. This assumes static datasets, which limits applicability in real-time, federated, or continually updated data environments. Please clarify whether you envision an extension of B3 to handle dynamic or growing datasets. Could B3 support periodic re-mining or incremental clustering, or might online approximation variants be viable?
2.	How sensitive is B3’s performance to the teacher model’s quality and alignment with the downstream task distribution? The success of B3 heavily depends on the accuracy of teacher-based similarity rankings. The paper briefly evaluates a stronger teacher but reports negligible gains, which seems counterintuitive. Please elaborate: Were the teacher models pretrained on the same modalities and tasks as the downstream evaluation tasks? Would using a mismatched or weak teacher degrade B3’s performance?
3.	Why was METIS selected as the clustering method, and how does it compare to alternative strategies like K-Means or spectral clustering in this context? The METIS algorithm is efficient and suits graph partitioning, but alternative clustering methods might offer different trade-offs (e.g., capturing latent structures, scalability, etc.). Please justify the choice of METIS beyond runtime. Could performance be further improved using embedding-based clustering techniques or data-driven adaptive clustering?
4.	Could you elaborate on the trade-off between cluster size (K) and performance, especially across diverse task types? You mention that K needs empirical tuning, and that high values degrade performance. However, it’s unclear whether this holds uniformly across retrieval, classification, and VQA tasks. Please clarify if K was tuned per task or globally. Does B3 generalize well with a fixed K across task types, or is task-specific tuning essential?

**Ethical Concerns:**

["NO or VERY MINOR ethics concerns only"]

**Limitations:**

see above

**Quality:**

3

**Strengths And Weaknesses:**

Strengths
1.	The new batch construction strategy is simple and effective. The B3 method is rigorously formulated, leveraging graph-based community detection (METIS) over teacher-model rankings to construct contrastive batches, and supported with theoretical bounds on InfoNCE loss approximation.
2.	Extensive experiments on the MMEB benchmark across 36 tasks and multiple model scales (2B and 7B) show consistent and substantial gains, especially under small batch sizes.
3.	While hard negative mining is not new, the use of graph-based community detection over ranked examples to construct entire informative batches is a novel framing.
Weakness
1.	The approach assumes that the teacher model produces semantically meaningful ranks. If the teacher is weak or biased, the batch quality may degrade significantly, affecting downstream performance.
2.	B3 assumes access to the full training dataset for batch construction. This makes it less suitable for continual learning or privacy-sensitive domains.
3.	B3 is more like a training technique that enhances contrastive learning by constructing more reliable negative samples. Can this technique be extended to more general tasks? In the paper, the method is primarily validated on the MMEB dataset. Despite MMEB's diversity, additional benchmarks, such as MTEB (text-only) or domain-specific tasks (e.g., medical imaging), could have further strengthened the generalizability claim.

---

> ### Author Rebuttal · Authors · 2025-07-30
>
> Thank you for your thoughtful comments and time reviewing our work. We are truly pleased with the positive reception our work has garnered from you and the other reviewers.
> * “method gives new SOTA performance…method is simple…should yield richer gradients…” - RM3oh
> * “B3 method is rigorously formulated…supported with theoretical bounds on InfoNCE loss approximation…consistent and substantial gains, especially under small batch sizes…graph-based community detection is a novel framing ” - RipHG
> * “has clear intuition and solid theoretical justification…Experiments seem solid and extensive…” - Rm62i
> * “achieves sota performance despite training at smaller batch size…ablation studies validate the effectiveness of B3” - R4DdK
>
> We equally appreciate your constructive comments, and have modified the draft to address the raised concerns.
>
> ---
> ---
>
> >B3 is more like a training technique that enhances contrastive learning by constructing more reliable negative samples. Can this technique be extended to more general tasks?....
>
> We concur with the reviewer that B3 is a highly generalizable method for contrastive training. To further substantiate our claim, we report the performance of B3, trained on the same NLI data as SimCSE (both with and without annotated hard negatives (HN)), evaluated on the STS benchmark. The corresponding SimCSE models (with/without HN) serve as the teacher models. All models (teacher and student) in the following table are 300M Roberta-Large.
>
>
> | Model   |  |  Teacher | | Data  | \|B\| |  | |STS12 | STS13 | STS14 | STS15 | STS16 | STSB | SICKR | Avg.   |
> |:--|:--:|:---|--|:---:|:--:|:--:|:--:|:---:|:---:|:--:|:--:|:---:|:----:|:-:|:---:|
> | SimCSE (w/o HN) | | - | | 275k | 512  | | |75.63 | 84.40 | 78.06 | 84.82 | 81.87 | 84.21 | 77.00 | 80.86  |
> | **B3 (w/o HN)** | | SimCSE (w/o HN) | | 275k | 512 | | | 77.49 | 85.73 | 80.37 | 85.36 | 82.75 | 84.24  | 75.40 | **81.62** |
> ||||||||||||||||
> | SimCSE | | - | | 275k | 512  | | | 77.30 | 86.68 | 82.20 | 86.58 | 84.02 | 86.36 | 81.84  | 83.57  |
> | **B3** | | SimCSE  | | 275k | 512  | | | 78.70 | 87.58 | 83.47 | 87.22 | 84.50 | 86.85 | 81.33  | **84.24** |
> ||||||||||||||||
>
>
> **B3 demonstrates notable improvements over SimCSE on the STS benchmark**, validating our batch mining approach for text-only tasks. Note that even starting with HN in the teacher, we can still see improvements. In future work, we plan to extend this to larger benchmarks like MMEB.
>
> ---
> ---
>
> >B3 assumes access to the full training dataset for batch construction. This makes it less suitable for continual learning or privacy-sensitive domains.
>
> - Contrastive learning typically necessitates substantial data volumes and large batch sizes, and is therefore **predominantly conducted in an offline setting**.
> - **Every single baseline** reported in Table 1 of the draft, as well as those on the MMEB leaderboard, operate in an offline configuration.
> - Likewise, **most leading models on the MTEB leaderboard**, including SFR‑Embed and NV‑Embed, are trained in an offline manner.
>
> We regard this not as a limitation but as a natural consequence of the extensive data and batch requirements inherent to contrastive learning. With regards to privacy, B3 is a training algorithm and requires access to the data like every other training method.
>
> Rui Meng, et al. 2024. Sfr-embedding-mistral: Enhance text retrieval with transfer learning.
>
> Gabriel de Souza et al. 2024, Nv-retriever: Improving text embedding models with effective hard-negative
> mining.
>
> ---
> ---
>
> >Can B3 adapt to dynamic or evolving datasets, such as in online or streaming learning scenarios? B3 relies on full-dataset ranking and offline clustering…might online approximation variants be viable?
>
> Online learning comprises two essential components: **(1) generating effective gradients from incoming examples and (2) retaining knowledge acquired from previously trained examples.**
> - For the latter, several approaches have been proposed; for instance, Liu et al. (2025) introduced a technique in which gradients from incoming batches are projected to prevent interference with existing knowledge.
> - For the former, B3 can be adapted to operate on a **sliding window of incoming examples**. Notably, at any given step, we already have a trained model available, which can serve as a teacher—**eliminating the need for an additional model—making B3 well-suited for this setting**.
>
> Liu, X. et al. (2025). Continual multimodal contrastive learning. arXiv preprint arXiv:2503.14963.
>
> ---
> ---
>
> >Why was METIS selected as the clustering method, and how does it compare to alternative strategies like K-Means or spectral clustering in this context? …. Please justify the choice of METIS beyond runtime...?
>
> Our choice of METIS algorithm is due to three main reasons.
> - Note that we need the **cluster size $K$ to be fixed** in B3 (as the batch size $|B|$ is fixed). Most clustering algorithms, such as K-Means or Agglomerative, have balanced variants, but these are typically much more computationally expensive, often running in $O(n^3)$. In contrast, METIS operates in $O(n \log⁡ n)$ while producing equal-sized clusters.
> - A key idea within B3 cluster construction is to **minimize the presence of false negatives (upto rank $p$)** in the training clusters. Note that each example in the training dataset has a different set of false negatives. Clustering algorithms like K-Means/Agglomerative would require substantial modifications to achieve this.
> - Examples with **ranks beyond $p+m$ in B3 are less relevant** and should not play a role in clustering. Here again, it is non-trivial to include these constraints in regular clustering algorithms.
>
> To accommodate rank-based constraints, we opted for graph-based clustering algorithms. Furthermore, to satisfy the fixed batch size requirement, we selected the METIS algorithm.
>
> ---
> ---
>
>  >Could you elaborate on the trade-off between cluster size (K) and performance, especially across diverse task types? …Does B3 generalize well with a fixed K across task types, or is task-specific tuning essential?
>
> - Saunshi et al. (2019) theoretically demonstrated that using an excessive number of negatives leads to intra-class collisions, which **degrades the effectiveness of contrastive learning in general—not limited to any specific domain such as retrieval or VQA**. They further validated this empirically across diverse image and text datasets. Similarly, SFR‑Embed reported that an excessive number of hard negatives can negatively impact performance on MTEB.
> - Accordingly, for B3—which yields $K$ strong negatives per example—we tuned the value of $K$ instead of simply selecting the maximum possible value (i.e., $K$=batch size). **A single $K$ value was used across all tasks, including retrieval, VQA, classification, and grounding**.
> - It is important to note that the batch size is a multiple of $K$, which prevents adjusting it individually for different datasets under our fixed‑batch‑size setup. As shown in the Table 1 and 4, a single $K$ value generalized for all datasets.
>
> Saunshi, et al. (2019, May). A theoretical analysis of contrastive unsupervised representation learning. In ICML. PMLR.
>
> ---
> ---
>
> >The approach assumes that the teacher model produces semantically meaningful ranks. If the teacher is weak or biased,... degrade significantly, affecting downstream performance. ...The success of B3 heavily depends on the accuracy of teacher-based similarity rankings. ...Were the teacher models pretrained on the same modalities and tasks as the downstream evaluation tasks? Would using a mismatched or weak teacher degrade B3’s performance?.
>
> In B3, the teacher’s sole role is to spot potential strong negatives and group them in the same batch. We simply take ranks $p:(p+m)$ from the teacher to build batches with strong in-batch negatives— and even with as few as 8 such negatives ($K=8$ in the Table), performance remains solid. Critically, the **teacher’s task is limited: it only needs to pick out some strong negatives, not all of them**.
>
> Hence, this **doesn’t require a very accurate teacher**, as the reviewer suspected. To prove the point, we run B3 with a much weaker teacher (CLIP). Despite CLIP performing far below VLM2Vec on MMEB, using it as a teacher still delivers strong results.
>
>
> | Model           |      | Teacher |  | \|B\|  | | K  |  |  | **Ret.** | **Cla.** | **VQA** | **Gro.** | **ID** | **OOD** | **Avg** |
> |---------------|---|---|---|----|---|--|---|--|-----------|-----------|-----------|-----------|------------|-------------|---------|
> | Random Batches (2B) | | - | | 1024 | | - | | |66.33 | 66.47|59.08|75.30|69.59|60.05|65.35|
> | B3 (2B) | | VLM2Vec2B | | 512 | | 8| | |69.81|66.94|59.93|80.00|71.01|62.89|67.40|
> | B3 (2B) | | VLM2Vec2B | | 512 | | 32  | | | 70.38     | 66.10     | 60.71     | 80.03     | 70.90      | 63.43       | 67.58   |
> | B3 (2B) | | VLM2Vec7B | | 512 | | 32 | | | 70.36     | 66.60     | 61.08     | 77.83     | 71.46      | 62.70       | 67.57   |
> | B3 (2B) | | **CLIP400M** | | 512 | | 32 | | | 69.79|66.34| 60.48     | 79.70     | 71.27      | 62.45       | **67.30**   |
> ||||||||||
>
>
> All that said, a strong teacher model can still be trained using only positives and a sufficiently large batch size, without relying on any negatives. VLM2Vec teacher models were trained solely with positives and random batches, without incorporating hard negatives. VLM2Vec models used in the paper were trained with the MMEB train set containing the same modalities as the MMEB eval set.
>
> ---
> ---
>
> Finally, we sincerely appreciate the reviewer’s thoughtful and insightful questions, which have helped us clarify and strengthen our work. We will incorporate the additional results, analyses, and detailed responses provided here into the final draft to ensure a more comprehensive and rigorous presentation.

---

> ### Comment · Area_Chair_WD3N · 2025-08-05
>
> Reviewer ipHG, please engage in the discussion period. I understand that the review period started over a weekend, but we only have a few days remaining in the (now slightly extended) discussion period. The authors have provided a thoughtful response to your review and you are obligated to respond to it. You should share with the authors if they addressed questions or concerns you had, and seek clarification about any questions or concerns that remain. Please post your response as soon as you can so that there is time for the authors to follow up and discussion to progress as needed.

---

> > ### Author Response · Authors · 2025-08-09
> > **Reminder: Response Deadline in a Few Hours**
> >
> > This is a gentle reminder that the author-reviewer discussion deadline is only a few hours away. We would appreciate it if you could let us know whether the rebuttal has addressed your concerns. We’re happy to address any additional questions you may have.
> >
> > We appreciate your time and consideration.

---

### Official Review · Reviewer_M3oh · 2025-07-08

**Clarity:** 3
**Significance:** 3
**Originality:** 2
**Rating:** 5
**Confidence:** 3

**Summary:**

The authors propose a new technique, B3, for assembling batches in a contrastive learning setup.  They find that by selecting more informative negative examples (through their B3 method), the performance of the models improve in the multimodal evaluation setting they consider.

**Questions:**

Why did you decide to anthropomorphize the model in Fig. 1?

**Ethical Concerns:**

["NO or VERY MINOR ethics concerns only"]

**Final Justification:**

I appreciate the additional experiments in the rebuttal as well as the comment about model capability in the teacher network. I think that this technique can be used to improve self supervised learning due to its broad applicability.

**Limitations:**

yes

**Paper Formatting Concerns:**

Minor: table captions should come before the table itself.

Minor: ll. 58 to 65: the list is not formatted properly; sota is an abbreviation/acronym, so should be capitalized.

**Quality:**

3

**Strengths And Weaknesses:**

# Strength

- The method gives new SOTA performance on multimodal evaluations (NB: I am not an expert in literature and do not know much related work in this domain.  I hope that other reivewers can assess this point better.)
- The method is simple, which is a strength
- The method can be applied in various domains, as it enhances the standard contrastive learning paradigm
- The initial idea appears sensible; by making the discrimination between positive and negative samples harder, the loss optimization should yield richer gradients.
- By sharding the training set into clusters, this could also give rise to distributed computation, thus improving the parallelism.
- The idea is also supported by other works, for example "The Optimal Noise in Noise-Contrastive Learning Is Not What You Think" (Chehab et al 2022).

# Weaknesses

As the evaluation is very costly, it is understandable that the authors only perform limited evaluations.  Nevertheless, I would have liked to see more ablations.  Especially in the construction of the matrix $S$ there are two hyperparameters involved, which are finetuned for the task itself.  I would have liked to see some study over a parameter range for both $p$ and $m$.  And do you see a reason for why VQA tasks benefit from a larger $p$?  I would also really like to see what happens when $p=0$.  Does the performance degrade because now you will sample images that are *too* similar?

The method relies on a pretrained model that accurately assigns similarity to the training dataset. Through this, the model will probably pick up features that the teacher network learned more quickly or strongly.  Is this something you can comment on?

Conversely to the previous paragraph:  can the method be reformulated to be completely self-supervised?  You could start with a randomly initialized teacher network and then throughout training update the network weights and recompute the communities which will be used to form batches.

One omission is maybe a simpler case.  How does B3 perform on for example standard CL image classification?  This should be comparatively easy to test and would give additional credence to the method.

---

> ### Author Rebuttal · Authors · 2025-07-30
>
> Thank you for your thoughtful comments and time reviewing our work. We are truly pleased with the positive reception our work has garnered from you and the other reviewers.
> - “method gives new SOTA performance…method is simple…should yield richer gradients…” - RM3oh
> - “B3 method is rigorously formulated…supported with theoretical bounds on InfoNCE loss approximation…consistent and substantial gains, especially under small batch sizes…graph-based community detection is a novel framing ” - RipHG
> - “has clear intuition and solid theoretical justification…Experiments seem solid and extensive…” - Rm62i
> - “achieves sota performance despite training at smaller batch size…ablation studies validate the effectiveness of B3”-R4DdK
>
> We equally appreciate your constructive comments, and have modified the draft to address the raised concerns.
>
> ---
> ---
>
> Regarding,
>
> >... reason for why VQA tasks benefit from a larger p?
>
> $p$ serves as a mechanism for excluding false negatives, a strategy previously employed in SFR‑Embed and NV‑Embed. For retrieval, we adopted the same value of $p=30$ as used in SFR‑Embed. NV‑Embed also showed that lower values of $p$ between $10$ and $30$ perform well. However, in the case of VQA training datasets in MMEB, the positive (target) responses were notably short. The table below reports the average target lengths for the VQA tasks.
>
> | Dataset | OK-VQA | A-OKVQA | DocVQA | InfoVQA | ChartQA | Visual7W | Avg.  |
> |:--|:--:|:--:|:--:|:----:|:---:|:--:|:--:|
> | Length | 1.24 | 1.28 | 2.16   | 1.61 | 1.12 | 1.98 | 1.57 |
> |||||||||
>
> In contrast, for image2text retrieval tasks, the average positive (target) length was as follows
> | Dataset | VisualNews_i2t | MSCOCO_i2t | WebQA | Avg. |
> |:--|:--:|:--:|:--:|:--:|
> | Length | 18.89 | 10.44 | 23.45 | 17.6 |
> ||||||
>
> Due to this reason, **we found a significantly higher number of targets which had similar/same meaning as that of the labelled target**. For instance, here is the first example in OK-VQA dataset. As seen, due to the short nature of the answers a higher number of targets of other examples are very similar/same compared to retrieval.
> - Image: Image containing a women playing tennis
> - Q: What is the hairstyle of the blond called?
> - Top ranked targets: [ponytail, ponytail, ponytail, pony tail, pony tail,...]
>
> Hence, we used a higher value of $p$ for VQA tasks. For computational efficiency, we tuned $p$ for smaller batch sizes and retained these values for larger batch sizes. Here are results for a model trained and tested on VQA datasets only at batch size ($|B|=32$) for different $p$ values.
>
> | p | \|B\|  | ID | OOD | Avg.  |
> |:--|:--:|:--:|:--:|:--:|
> | 30|32| 57.80 | 56.73 | 57.37 |
> | 70|32| 58.38 | 57.95 | 58.21 |
> ||||||
>
> At lower batch sizes, results improve for higher $p$ values. However, as discussed next, at higher batch sizes, $p$ has a smaller effect on the performance.
>
> ---
>
> >... see some study over a parameter range for both p and m
>
> We hereby present some analysis on parameter $m$. Model trained and tested on retrieval datasets only at batch size 32.
>
> | p | m | \|B\| | ID | OOD | Avg.|
> |:--|:--:|:--:|:--:|:--:|:--:|
> | 30 | 100 | 32 | 72.83 | 55.50 | 67.05 |
> | 30 | 500 | 32 | 72.25 | 54.65 | 66.38 |
> | 30 | 800 | 32 | 72.60 | 55.15 | 66.78 |
> |||||||
>
> At smaller batch sizes, the performance changes with $m$. $m=100$ as used in SFR-Embed worked well for us. We demonstrate the ablations over $p$ in the next question.
>
> ---
> >... what happens when p=0.
>
> The table below reports results for $p=0$ across different batch sizes. At smaller batch sizes, using an elevated value of $p$ proves beneficial, effectively reducing multiple false negatives. However, at larger batch sizes, this effect diminishes; as shown, **performance remains comparable between $p=0$ and $p=30$ when using higher batch sizes.**
>
> | p (Ret,VQA) | p+m (Ret, VQA) |  |   | \|B\|  | ID | OOD  |  Avg.  |
> |:--|:--:|:--:|:--:|:--:|:--:|:--:|:--:|
> | 0,0  | 30,70  | | |32   | 62.18  | 58.23  | 60.42  |
> | 30,70 | 130,170| | | 32  | 63.44  | 59.67  | 61.76  |
> | 0,0  | 30,70 | | |128 | 68.16 | 61.99  | 65.42  |
> | 30,70 | 130,170| | | 128 | 68.35  | 61.80  | 65.44  |
> | 0,0 | 30,70  | | | 512 | 71.09 | 63.31 | 67.63 |
> | 30,70 | 130,170 | | | 512 | 70.90 | 63.43 | 67.58 |
> |||||||||
>
> Presented below are additional variations of $p$ and $m$ at a higher batch size (512). As shown, **B3 maintains consistent performance across these variations when using a larger batch size.**
>
> | p (Ret, VQA)  | p+m (Ret, VQA) | | | \|B\|  | ID | OOD | Avg. |
> |:--|:--:|:--:|:--:|:--:|:--:|:--:|:--:|
> | 0,0|30,70| | | 512 | 71.09 | 63.31 | 67.63 |
> | 0,0|100,100  | | | 512 | 71.06 | 63.24 | 67.59 |
> | 30,70 | 130,170 | | | 512 | 70.90 | 63.43 | 67.58 |
> |||||||||
>
> As shown $p=0$ works very well at larger batch sizes.
>
> **Main Takeaway (from $p$ and $m$ analysis)**
> - *The previous tables indicate that the hyperparameters $p$ and $m$ have a greater impact on performance at smaller batch sizes. While B3 requires tuning of these parameters to achieve optimal results at smaller batch sizes, it shows increased robustness to their variation at larger batch sizes.*
>
> ---
> ---
>
> >... can the method be reformulated to be completely self-supervised? You could start with a randomly initialized teacher network ...
>
> - Contrastive learning relies on two key signals: positives and negatives.
> - Our method specifically **focuses on selecting negatives in a self‑supervised manner, given a set of positives.**
> - Constructing a pipeline of progressively harder negatives, as suggested by the reviewer, is feasible, but we leave this for future exploration.
> - **Self‑supervising positives** is an orthogonal direction, and B3 can be seamlessly **integrated with existing approaches such as BYOL, DINOv2, or V‑JEPA** that address this aspect.
>
> ---
> ---
>
> >... How does B3 perform on for example standard CL image classification? … additional credence to the method.
>
> We are not sure what the reviewer means by "standard CL image classification", but here are zero-short ImageNet1k classification results. MMEB benchmark in Table 1 additionally contains 10 different classification tasks.
>
> | Model | | | |ImageNet1k |
> |:--|--|--|--|:--:|
> | B3 Qwen7B | | | | **84.1** |
> | B3 Qwen2B  | | | | **83.2** |
> | VLM2Vec Qwen7B | | | | 80.5 |
> | VLM2Vec Qwen2B | | | | 77.5 |
> | CLIP  | | | | 55.8 |
> ||||||
>
> We are happy to provide any further results the reviewer may wish to see.
>
> ---
> ---
> >The method relies on a pretrained model that accurately assigns similarity to the training dataset. …. Is this something you can comment on?
>
> In B3, the teacher’s sole role is to spot potential strong negatives and group them in the same batch. We simply take ranks $p:(p+m)$ from the teacher to build batches with strong in-batch negatives— and even with as few as 8 such negatives ($K=8$ in Table 4), performance remains solid. Critically, the **teacher’s task is limited: it only needs to pick out some strong negatives, not all of them**.
>
> Hence, this **doesn’t require a very accurate teacher**, as the reviewer suspected. To prove the point, we run B3 with a much weaker teacher (CLIP). Despite CLIP performing far below VLM2Vec on MMEB, using it as a teacher still delivers strong results.
>
>
> | Model | | Teacher | | \|B\|  | | K | | | **Ret.** | **Cla.** | **VQA** | **Gro.** | **ID** | **OOD** | **Avg** |
> |:--|--|:---:|---|:--:|:---:|:--:|:---:|:--:|:---:|:---:|:----:|:---:|:---:|:---:|:--:|
> | Random Batches (2B) | | - | | 1024 | | - | | |66.33 | 66.47|59.08|75.30|69.59|60.05|65.35|
> | B3 (2B) | | VLM2Vec2B | | 512 | | 8| | |69.81|66.94|59.93|80.00|71.01|62.89|67.40|
> | B3 (2B) | | VLM2Vec2B | | 512 | | 32  | | | 70.38  | 66.10 | 60.71 | 80.03 | 70.90 | 63.43 | 67.58  |
> | B3 (2B) | | VLM2Vec7B | | 512 | | 32 | | | 70.36 | 66.60  | 61.08 | 77.83  | 71.46 | 62.70 | 67.57 |
> | B3 (2B) | | **CLIP400M** | | 512 | | 32 | | | 69.79|66.34| 60.48 | 79.70 | 71.27 | 62.45 | **67.30** |
> ||||||||||
>
>
> All that said, a strong teacher model can still be trained using only positives and a sufficiently large batch size, without relying on any negatives. VLM2Vec teacher models were trained solely with positives and random batches, without incorporating hard negatives.
>
> ---
> ---
>
> >The method can be applied in various domains, as it enhances the standard contrastive learning paradigm
>
>
> We concur with the reviewer that B3 is a highly generalizable method for contrastive training. To further substantiate our claim, we report the performance of B3, trained on the same NLI data as SimCSE (both with and without annotated hard negatives (HN)), evaluated on the STS benchmark. The corresponding SimCSE models (with/without HN) serve as the teacher models. All models (teacher and student) in the following table are 300M Roberta-Large.
>
>
> | Model |  | Teacher | | Data | \|B\| | | |STS12 | STS13 | STS14 | STS15 | STS16 | STSB | SICKR | Avg. |
> |:--|:--:|:--|--|:--:|:--:|:--:|:--:|:--:|:---:|:--:|:--:|:--:|:--:|:-:|:--:|
> | SimCSE (w/o HN) | | - | | 275k | 512  | | |75.63 | 84.40 | 78.06 | 84.82 | 81.87 | 84.21 | 77.00 | 80.86  |
> | **B3 (w/o HN)** | | SimCSE (w/o HN) | | 275k | 512 | | | 77.49 | 85.73 | 80.37 | 85.36 | 82.75 | 84.24  | 75.40 | **81.62** |
> ||||||||||||||||
> | SimCSE | | - | | 275k | 512  | | | 77.30 | 86.68 | 82.20 | 86.58 | 84.02 | 86.36 | 81.84  | 83.57  |
> | **B3** | | SimCSE | | 275k | 512  | | | 78.70 | 87.58 | 83.47 | 87.22 | 84.50 | 86.85 | 81.33  | **84.24** |
> ||||||||||||||||
>
>
> **B3 demonstrates notable improvements over SimCSE on the STS benchmark**, validating our batch mining approach for text-only tasks. Note that even starting with HN in the teacher, we can still see improvements. In future work, we plan to extend this to larger benchmarks like MMEB.
>
>
> ---
> ---
>
> Finally, we sincerely appreciate the reviewer’s thoughtful and insightful questions, which have helped us clarify and strengthen our work. We will incorporate the additional results, analyses, and detailed responses provided here into the final draft.

---

> > ### Comment · Reviewer_M3oh · 2025-08-03
> >
> > I appreciate the thorough response and additional evaluation.  The comment with the capability of the teacher network is good, thank you for that.
> >
> > I would be happy to see this accepted with the promised changes and will change my score to reflect that.

---

> > > ### Author Response · Authors · 2025-08-03
> > > **Thank you**
> > >
> > > We are pleased that our rebuttal addressed your questions and contributed to an improved score. Your feedback has been extremely valuable to us.

---

### Note · Authors · 2025-08-11

Hi AC/SAC,

Thank you for your time and effort in reviewing our work. We also appreciate the diligent efforts of all reviewers. We hope you will consider the following points while drafting a meta review:

---

_**Summary of Core Contributions**:_
- Solid **theoretical foundation** underlying the proposed methodology.
- Achieves **state-of-the-art** results on the MMEB leaderboard.
- Surpasses the previous SOTA while using **4–16× smaller batch sizes**.
- **B3 without hard negatives** outperforms the Random Batching baseline with five strong hard negatives, while requiring **only half the compute and training time**.
- **B3 works without a strong teacher**. Even with a relatively weak teacher (e.g., CLIP), B3 yields substantial performance gains.
- **Generalizable across contrastive learning models and domains**, with notable improvements observed on STS-style, text-only tasks.

---

_**Positive Reviews from ALL Reviewers**_
- “method gives new SOTA performance…method is simple…should yield richer gradients…” - RM3oh
- “B3 method is rigorously formulated…supported with theoretical bounds on InfoNCE loss approximation…consistent and substantial gains, especially under small batch sizes…graph-based community detection is a novel framing ” - RipHG
- “has clear intuition and solid theoretical justification…Experiments seem solid and extensive…” - Rm62i
- “achieves sota performance despite training at smaller batch size…ablation studies validate the effectiveness of B3” - R4DdK

---

_**Concerns about Rebuttal**:_
- No response to rebuttal from Reviewer ipHG
  - We've addressed every concern from the review in the rebuttal. In most cases, we provided an **in-depth quantitative analysis** to support our rebuttal claims. We would greatly appreciate if Reviewer ipHG could take a moment to review our rebuttal and consider the strength of our contributions when making any final updates.

---

We believe that this work represents a significant finding that should be shared with the community to advance future contrastive training practices. The **principles developed in this work can be directly applied to leading MTEB embedding models** such as Gemini-Embedding, Qwen3-Embed, SFR-Embed, NV-Embed, and others.

We request that you consider the context and significance of the main contributions when evaluating the reviews/rebuttal and reaching the final decision.

---

### Decision · Program_Chairs · 2025-09-17

**Decision:**

Accept (spotlight)

**Comment:**

The paper introduces B3, a method for improving contrastive learning by constructing batches enriched with high-quality in-batch negatives using a graph-based mining approach. The method forms informative training batches, which when trained upon outperform existing methods across various batch sizes and model scales.

Reviewers consistently praised the method's conceptual simplicity, strong empirical results, and novel use of community detection for batch construction. They noted the SOTA performance on the MMEB benchmark, the effectiveness at small batch sizes, broad applicability across domains, and robust performance even with weaker teacher models as significant positives. Reviewers also appreciated that the paper included theoretical analysis and extensive ablations validating the method’s components.

One reviewer (ipHG) gave a borderline rating and raised concerns about generalizability beyond MMEB, sensitivity to teacher model quality, and the method’s reliance on full offline access to the dataset. However, these concerns seem to have been sufficiently addressed in the rebuttal. While ipHG did not engage in discussion, the AC was compelled by the author response and does not believe that substantive objections remained unaddressed.

For the camera-ready version, the authors may consider clarifying the terminology around cross-modal positive pairs and elaborating on the rationale behind tuning hyperparameter, p, for different task types, as discussed in response to Reviewer 4DdK.

The AC recommends acceptance.